# Molecular mechanics underlying flat-to-round membrane budding in live secretory cells

Wonchul Shin[1,5], Ben Zucker [2,5], Nidhi Kundu[3,5], Sung Hoon Lee[4], Bo Shi[1], Chung Yu Chan [1], Xiaoli Guo[1], Jonathan T. Harrison[3], Jaymie Moore Turechek[1], Jenny E. Hinshaw[3✉], Michael M. Kozlov [2✉] & Ling-Gang Wu [1✉]

Membrane budding entails forces to transform flat membrane into vesicles essential for cell survival. Accumulated studies have identified coat-proteins (e.g., clathrin) as potential budding factors. However, forces mediating many non-coated membrane buddings remain unclear. By visualizing proteins in mediating endocytic budding in live neuroendocrine cells, performing in vitro protein reconstitution and physical modeling, we discovered how non-coated-membrane budding is mediated: actin filaments and dynamin generate a pulling force transforming flat membrane into Λ-shape; subsequently, dynamin helices surround and constrict Λ-profile's base, transforming Λ- to Ω-profile, and then constrict Ω-profile's pore, converting Ω-profiles to vesicles. These mechanisms control budding speed, vesicle size and number, generating diverse endocytic modes differing in these parameters. Their impact is widespread beyond secretory cells, as the unexpectedly powerful functions of dynamin and actin, previously thought to mediate fission and overcome tension, respectively, may contribute to many dynamin/actin-dependent non-coated-membrane buddings, coated-membrane buddings, and other membrane remodeling processes.

[1] National Institute of Neurological Disorders and Stroke, Bethesda, MD, USA. [2] Department of Physiology and Pharmacology, Sackler Faculty of Medicine, Tel Aviv University, 69978 Ramat Aviv, Israel. [3] Structural Cell Biology Section, Laboratory of Cell and Molecular Biology, National Institute of Diabetes and Digestive and Kidney Diseases, Bethesda, MD, USA. [4] Chung-Ang University, 84 Heukseok-ro, Dongjak-gu, Seoul 06974, Republic of Korea. [5]These authors contributed equally: Wonchul Shin, Ben Zucker, Nidhi Kundu. ✉email: jennyh@niddk.nih.gov; michk@tauex.tau.ac.il; wul@ninds.nih.gov

Membrane budding entails molecular forces to transform flat membrane into oval/round vesicles, which mediates many fundamental processes, such as endocytosis, intracellular trafficking, and virus infection[1]. Decades of studies suggest that the budding force may arise in part from ~40–120 nm cage-like structures coating the budding vesicle and formed by multimerization of clathrin, COPII, or COPI[1–3]. However, many budding processes are not associated with, and thus do not rely on these core coat-proteins. These non-coated-membrane budding processes mediate many biological processes, such as exosome budding for cell communications, virus budding for viral infection (e.g., Influenza and HIV virus), generation of hormone/transmitter-laden granules much larger than coated vesicles, and vesicle budding for clathrin-independent endocytosis[1,4,5]. Vesicle budding for clathrin-independent endocytosis mediates cellular uptake of many extracellular ligands, receptors, viruses (e.g., Ebola, HIV, Lassa, Herpes, Dengue, and SV40 viruses), bacteria, prions, and bacterial toxins (e.g., cholera and Shiga toxins)[5–7]. In the nervous and endocrine system, clathrin-independent vesicle budding maintains exocytosis and synaptic transmission by mediating most forms of endocytosis, including ultrafast endocytosis (<~0.1 s), fast endocytosis (<~3 s), bulk endocytosis of endosome-like vesicles, and even slow endocytosis (~10–60 s) previously assumed to be clathrin-dependent[8–13]. In brief, non-coated-membrane budding plays crucial physiological and pathological roles.

What type of physical forces mediate formation of non-coated vesicles? Many studies suggest that cytoskeletal filamentous actin (F-actin) is involved[5–7]. For example, inhibition of actin polymerization impairs several forms of clathrin-independent endocytosis, such as ultrafast endocytosis[10,14] and fast endophilin-mediated endocytosis[15] (for review, see refs. [5–7]). Deletion of actin β- or γ-isoform or application of latrunculin A that inhibits F-actin polymerization reduces the pit numbers during synaptic vesicle endocytosis that is considered clathrin-independent[11,14,16,17]. These results suggest actin involvement in non-clathrin-coated pit formation[5–7]. However, how F-actin contributes to the generation of the physical forces driving the non-coated vesicle budding remains unclear.

The study of non-coated-membrane budding is impeded by the difficulty of recognizing the intermediate structure in the flat-to-round transformation, which may implicate the potential force requirements. While electron microscopy (EM) is by far the only technique to see tiny structures, without core coat-proteins that generate an EM-recognizable dense protein coat, it is difficult to tell if a curved membrane configuration is an intermediate structure of budding, fusion, membrane folding, membrane movement, or an unknown shaping process. In fact, even for coated-membrane budding, the flat-to-round membrane dynamics remain, essentially, unclear in spite of numerous speculations based on EM of fixed samples that do not provide information about the potentially rapid and reversible processes[2,3]. It remains unsettled whether, and to what extent, core coat-proteins and non-coat-proteins contribute to coated-membrane budding[2,3]. Thus, the difficulty of real-time observing the budding process has become a bottleneck hindering us from understanding both non-coated- and coated-membrane budding.

Overcoming this technical bottleneck, a recent study uses super-resolution stimulated emission depletion (STED) microscopy to visualize in real-time the entire endocytosis membrane budding process in live neuroendocrine chromaffin cells, a common secretory cell model for the study of exo- and endocytosis[18]. This endocytic budding generates ~200–1500 nm vesicles[18], which are too large to be coated with clathrin or caveolin that forms ~40–100 nm cages[2]. Electron microscopy confirms that typical clathrin coatings are not observed at the budding intermediates or budding vesicles[18,19]. Thus, real-time endocytic budding without apparent coatings like clathrin can be observed in chromaffin cells. This budding process consists of three membrane transformation steps, flat- to Λ-shape (Flat→Λ), Λ- to Ω-shape (Λ→Ω), and Ω- to O-shape vesicle transition (Ω→O)[18]. By real-time imaging of these transitions, genetic and pharmacological manipulation aiming to identify proteins involved in these transitions, in vitro reconstitution of proteins' unconventional functions, and realistic physical modeling, we discovered two types of budding forces underlying the non-coated-membrane budding. First, a pulling force mediated by polymerized actin filaments together with the GTPase dynamin (well known as an enzyme mediating fission of ~5 nm narrow necks)[20,21] transforms a flat membrane into Λ-shape (Λ). Second, dynamin helices surround and constrict the Λ-profile's base, converting Λ into Ω-shape (Ω), and then constrict the Ω's pore until a vesicle is formed. Thus, dynamin and actin provide the pulling and constriction forces required for the flat-to-round transition to control the budding speed, vesicle size, and number, explaining how numerous endocytic modes differ in speed, vesicle size, and number can be generated[8,18,22]. The budding forces generated by dynamin and actin may contribute to mediate many non-coated-membrane buddings, coated-membrane buddings, and other membrane shaping processes (e.g., mitochondria division, cell migration) depending on dynamin (or dynamin-like proteins) and/or actin. Current models for dynamin/actin-dependent non-coated-membrane budding and coated-membrane budding may need to be re-examined and modified to account for the crucial role of dynamin and actin in Flat→Λ, Λ→Ω and Ω→O transitions.

## Results

**Experimental setting for observing Flat→Λ→Ω→O in real-time.** To observe membrane transformations, we overexpressed bovine adrenal chromaffin cells in primary culture with phospholipase C delta PH domain attached with mNeonGreen ($PH_G$), which binds to $PtdIns(4,5)P_2$ at, and thus labels the plasma membrane (Fig. 1a)[18,23,24]. With Atto 532 (A532, 30 μM) in the bath, we performed STED $PH_G$/A532 imaging every 26–200 ms per frame at the XZ plane with the Y-plane fixed at about the cell center ($XZ/Y_{fix}$ scanning; X: 15–20 μm, Z: 0.7–2.5 μm, Fig. 1a). Whole-cell 1 s depolarization ($depol_{1s}$, −80 to +10 mV) induced calcium currents (ICa), capacitance (Cm) changes reflecting exo-endocytosis (Fig. 1a)[25], fusion events (not analyzed here, but see ref. [24]), and endocytic membrane transitions (imaging duration: 60 s; 1 $depol_{1s}$ per cell, and 513 cells). Endocytic transitions were systematically described recently[18], which provided the foundation for the present work – the study of their underlying mechanisms. These transitions are summarized below to provide the background for the current study (for detail, see ref. [18]).

$Depol_{1s}$ induced transition of the flat membrane (Flat) through intermediate Λ-shape (Λ) and Ω-shape profile (Ω) to oval/round-shape vesicle (O), that is, Flat→Λ→Ω→O transition (Fig. 1b)[18]. This was the most complete, but not the only transition. $Depol_{1s}$-induced transitions along the Flat→Λ→Ω→O pathway could start from Flat, preformed Λ (pre-Λ, formed before $depol_{1s}$) or preformed Ω (pre-Ω, formed before $depol_{1s}$) and stop at Λ, Ω or O. These different combinations included Flat→Λ→Ω→O (Fig. 1b), Flat→Λ→Ω, Flat→Λ (Fig. 1c), pre-Λ→Ω→O, pre-Λ→Ω, and pre-Ω→O (see below, see also ref. [18] for detail). These transitions reflect three modular transitions, including Flat→Λ, Λ→Ω and Ω→O, which mediate the Flat-to-O transition. In the following, we presented new results describing how actin and dynamin are involved in mediating Flat→Λ; then we described how dynamin is involved in Λ→Ω and Ω→O, where

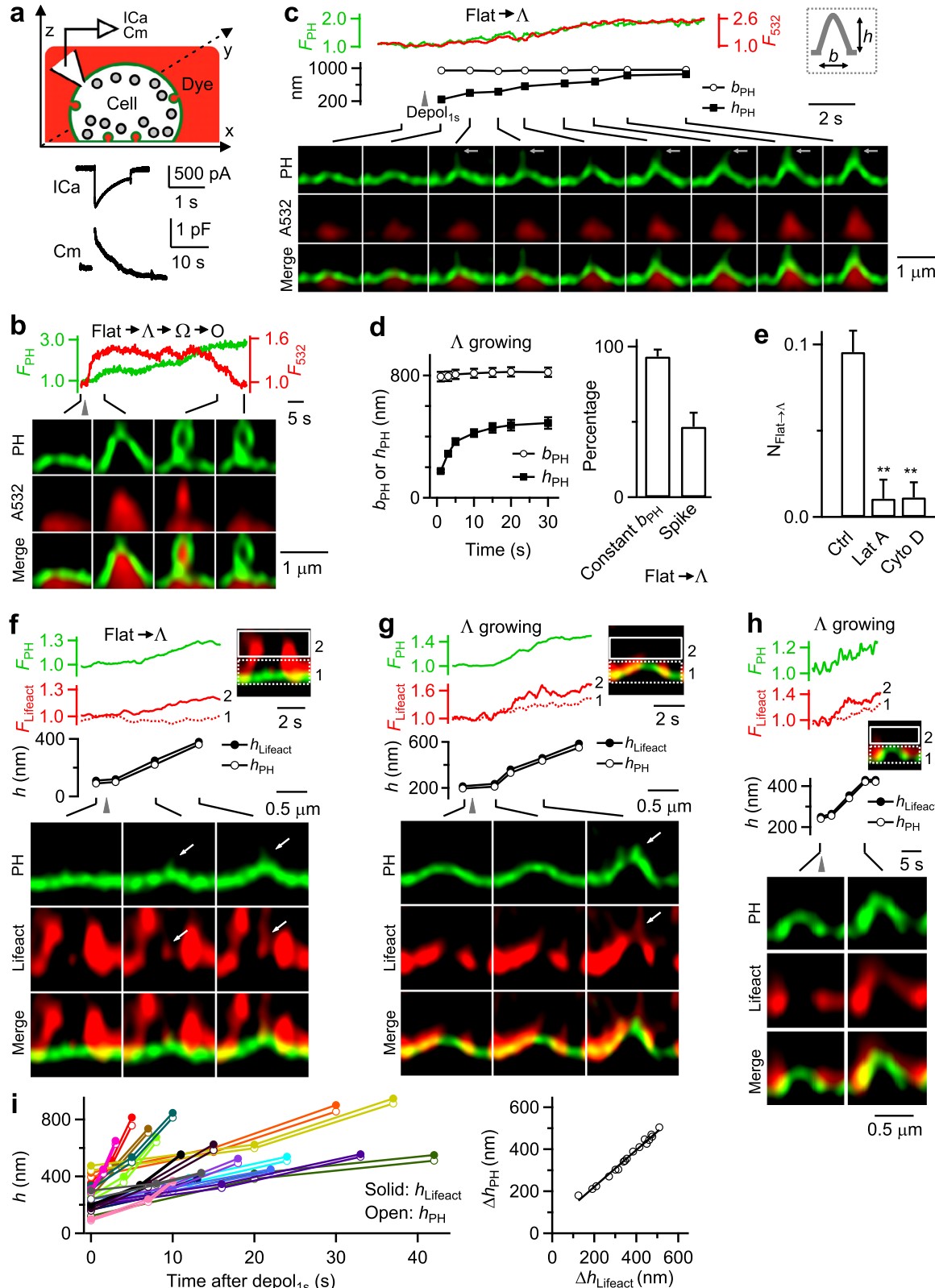

differentiation between Ω and O will be introduced. Finally, we performed realistic mathematical modeling to determine whether two forces, membrane pulling and constriction, are sufficient to mediate Flat-to-O transformation.

**Actin mediates Flat→Λ by pulling membrane inward.** During Flat→Λ, as the height of PH$_G$-labeled Λ ($h_{PH}$) increased, the base of

PH$_G$-labeled Λ ($b_{PH}$) remained unchanged at ~800 nm (Fig. 1c, d)[18]; thin tube-shape or spike-like protrusions of ~100–500 nm transiently appeared and disappeared at the growing Λ's tip (Fig. 1c, arrows, and Fig. 1d). The protrusions were not always observed (38 out of 79 Flat→Λ events, Fig. 1d), likely because they were short-lived, deviating from the fixed Y-plane (during XZ/Y$_{fix}$ scanning), and/or too small to be detected. These results suggest that a localized

**Fig. 1 Actin filaments in mediating Flat→Λ in chromaffin cells. a** Upper: setup drawing – a $PH_G$-labeled cell (green) in A532-containing bath (red) with whole-cell recording of calcium currents (ICa) and capacitance (Cm). x, y and z: microscopic axis. Lower: sampled ICa and Cm induced by $depol_{1s}$. **b** $PH_G$ fluorescence ($F_{PH}$, normalized to baseline), A532 fluorescence ($F_{532}$, normalized to baseline), and STED $XZ/Y_{fix}$ images (at times indicated with lines) showing Flat→Λ→Ω→O transition induced by $depol_{1s}$ (gray triangle). Distinguishing Ω from O is described Fig. 6 and the corresponding main text. **c** $F_{PH}$, $F_{532}$, $PH_G$-labeled Λ's height ($h_{PH}$) and base ($b_{PH}$, see also inset) and $XZ/Y_{fix}$ images (at times indicated with lines) showing Flat→Λ transition induced by $depol_{1s}$ (gray triangle). White arrows point to thin tube-shape protrusions that may be caused by a pulling force. **d** Left: Λ's $h_{PH}$ and $b_{PH}$ plotted versus time during Flat→Λ (mean ± SEM, 30 cells from 27 cultures; 54 bovines). $b_{PH}$ was not measured when $h_{PH}$ was <20% of the peak. Right: percentage of Flat→Λ transition with a constant base ($b_{PH}$ change <20%) or with thin tube-like protrusion at growing Λ's tip (Flat→Λ transitions; 30 cells from 27 cultures; 54 bovines). **e** The number of Flat→Λ transitions per 10 μm membrane along X-axis during $XZ/Y_{fix}$ scanning ($N_{Flat→Λ}$) in control (Ctrl, 513 cells from 168 cultures; 336 bovines), and in the presence of latrunculin A (Lat A, 3 μM, 20–30 min, 61 cells from 15 cultures; 30 bovines; $p = 0.005$), or Cytochalasin D (Cyto D, 4 μM, 20–30 min, 57 cells from 12 cultures; 24 bovines; $p = 0.006$). **$p < 0.01$ (two-tailed unpaired $t$-test, compared to Ctrl). Data are presented as mean + SEM. **f–h** $F_{PH}$, lifeact-mTFP1 fluorescence ($F_{lifeact}$), $PH_G$-labeled Λ's $h_{PH}$, the highest position of lifeact-mTFP1-labeled F-actin associated with Λ ($h_{lifeact}$), and sample $XZ/Y_{fix}$ images showing F-actin filament recruitment, attachment at, and movement with the growing Λ's tip during Flat→Λ (**f**) or growing of Λ (**g**, **h**). $F_{lifeact}$ from regions 1 (near Λ's base) and 2 (above Λ's base, inset) are plotted. **f**, **g** Spike-like protrusion attached to growing F-actin filaments (arrows). **h** F-actin association with Λ's tip, side and base. $h_{PH}$ and $h_{lifeact}$ were measured as the height from the $PH_G$-labeled base membrane. **i** Left: $h_{PH}$ and $h_{lifeact}$ (h) measured during Λ growing plotted versus time after $depol_{1s}$ (each colour: one Λ growing event). Right: the increase of $h_{PH}$ ($Δh_{PH}$) plotted versus the increase of $h_{lifeact}$ ($Δh_{lifeact}$, left) during Λ growing (each symbol: one Λ growing event). The line is a linear regression fit. Source data are provided as a Source Data file.

pulling force was applied to a boundary-confined membrane patch at the tip of the growing Λ−shape. This force directed towards the cytosol and mediating Flat→Λ-shape transition, apparently, varied in time.

Two sets of evidence suggest that F-actin is involved in mediating Flat→Λ. First, latrunculin A (61 cells) or cytochalasin D (57 cells), which specifically disrupted F-actin in chromaffin cells (e.g., Supplementary Fig. 1a) (for detail, see ref. [26]), substantially reduced the number of Flat→Λ ($N_{Flat→Λ}$) induced by $depol_{1s}$ (Fig. 1e), but not ICa or capacitance jumps (Supplementary Fig. 1b, c). Second, as Λ and its $h_{PH}$ increased, the fluorescence of lifeact-mTFP1-labeled F-actin ($F_{lifeact}$) grew stronger and appeared as a ~100–950 nm filament with one end attached to the growing Λ's tip (including the spike-like membrane protrusion above the tip), and the other end moving towards the cytosol (Fig. 1f–h, $n = 16$; Supplementary Movie S1; see also Supplementary Fig. 2 with less image contrast). Some filaments were >1 μm, but beyond image frame to measure. As $h_{PH}$ increased, the highest position of lifeact-mTFP1-labeled F-actin associated with Λ ($h_{lifeact}$) increased in parallel (Fig. 1f–i). F-actin fluorescence also increased at Λ's side and base (Fig. 1g–i). These results suggest that F-actin filament is recruited to attach at Λ, including at Λ's tip to pull membrane inward, hence, resulting in a growing Λ and membrane protrusion from Λ's tip.

**Gene knockout revealed actin involvement in Λ formation at synapses.** To determine whether actin's involvement in Λ formation applies to secretory cells beyond chromaffin cells (Fig. 1), we examined synapses where actin is crucial for all detectable forms of endocytosis[9,16,27]. We examined Λ and Ω formation with electron microscopy (EM) at mouse hippocampal cultures lacking β-actin, the main actin isoform crucial for endocytosis in the hippocampus[16]. To delete the gene encoding β-actin (Actb), we generated Cre-ER$^{TM}$; Actb$^{LoxP/LoxP}$ mice by crossing $CAGGCre$-$EM^{TM}$ mice, which contain tamoxifen-inducible cre-mediated recombination system in diverse tissues (Hayashi and McMahon, 2002), with Actb$^{LoxP/LoxP}$ mice (for detail, see ref. [16]). Hippocampal cultures from Cre-ER$^{TM}$; Actb$^{LoxP/LoxP}$ mice were treated with 4-OH-tamoxifen (TM, 1 μM). 4 days later, β-actin in the culture (TM4d-Actb$^{-/-}$ cultures) was reduced to 13 ± 7% of control ($n = 4$ western blots)[16]. To induce exo- and endocytosis, we applied 90 mM KCl for 1.5 min, and fixed samples at 0, 3, and 10 min after the end of KCl application. EM in wildtype boutons showed that compared to the resting condition, the number of Λ

and Ω per μm² of bouton cross-section (see Methods for definition) increased after KCl application and continued to increase 3–10 min later (Fig. 2a–c), reflecting endocytic pit generation. In TM4d-Actb$^{-/-}$ boutons, the number of Λ or Ω after KCl application was not increased and was much lower than the corresponding control at every time point measured (Fig. 2c). This observation, together with the previous finding that endocytosis is inhibited in TM4d-Actb$^{-/-}$ boutons[16], suggests that β-actin is involved in Λ formation at synapses. Since Λ and Ω size at synapses was less than 100 nm (Fig. 2a), much smaller than those observed in chromaffin cells (Fig. 1), actin involvement in Λ formation is thus applicable for different sizes ranging from <100 nm to ~1000 nm.

**Dynamin is also involved in mediating Flat→Λ.** Dynamin has been reported to interact with actin[20,28–30]. Here we present two sets of evidence suggesting that dynamin is also involved in mediating Flat→Λ in chromaffin cells. First, dynamin inhibitor dynasore (64 cells) or overexpressed dominant-negative mutant dynamin 1 K44A (106 cells) substantially reduced $N_{Flat→Λ}$ (Fig. 3a), but not ICa or capacitance jumps (Supplementary Fig. 3a, b). Second, STED $XZ/Y_{fix}$ imaging showed that when Λ was growing, dynamin 1-mTFP1 (or dynamin 2-mTFP1) fluorescence ($F_{dyn}$) increased at Λ's base, side, and tip, which paralleled the increase of $h_{PH}$; the highest position of dynamin 2-mTFP1 associated with Λ ($h_{dyn}$) also increased in parallel with $h_{PH}$ increase (Fig. 3b, c, 11 out of 11 events). These results suggest that dynamin is recruited to mediate Λ growing during Flat→Λ.

During Flat→Λ, Λ's base remained unchanged (Fig. 1c, d) and was surrounded with actin and dynamin scaffold (Figs. 1f–h and 3b). STED XZ-plane imaging every 50 nm along the Y-axis for 1-5 μm ($XZ/Y_{stack}$, Fig. 1a) confirmed that dynamin 1-mTFP1 (or dynamin 2-mTFP1) surrounded $PH_G$-labeled pre-Λ's base and side (Fig. 3d, e; 65 out of 67 Ω, 17 cells). Similarly, STED $XZ/Y_{stack}$ imaging showed that F-actin surrounded pre-Λ's base and side (Fig. 3f, g, 34 out of 34 Λ, 11 cells). These Λ-base-surrounding dynamin and actin scaffolds may serve as a physical barrier to prevent Λ's base from expansion during Flat→Λ, which may explain why Λ's base remained unchanged during Flat→Λ (Fig. 1c, d). Overall, these results (Figs. 1–3) strongly suggest that F-actin and dynamin co-mediate Flat→Λ by pulling membrane at the growing Λ's tip, and by surrounding and confining Λ's base from expansion (Fig. 3h).

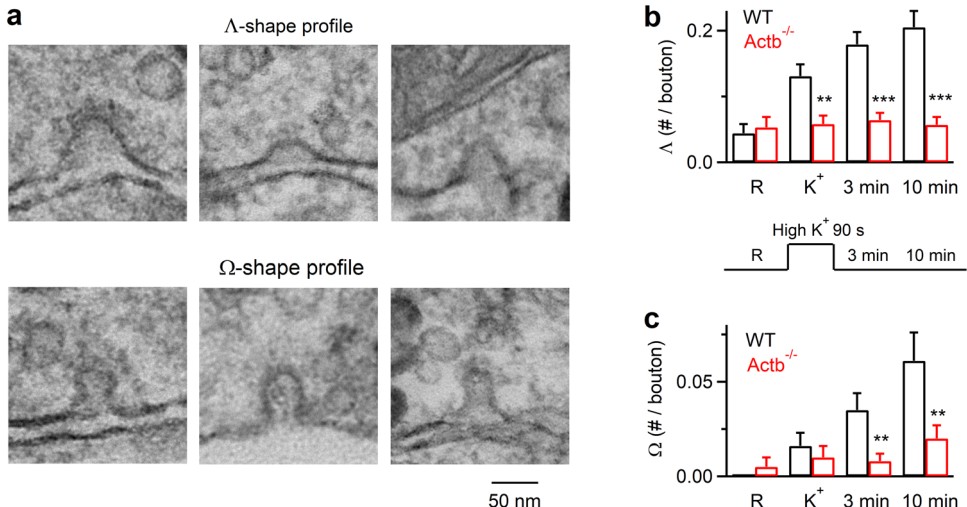

**Fig. 2 Reduction of Λ-profile formation in hippocampal boutons lacking β-actin. a** Sampled EM images showing the flat membrane, Λ-shape, and Ω-shape membrane at hippocampal boutons. The tip of Λ-shape profile could be either pointed or dome-like. **b** The number of Λ per bouton before (R) and after KCl application in wildtype (WT) or TM4d-Actb$^{-/-}$ synapses (mean + SEM; each group was from 40–100 synaptic profiles from 5–7 cultures; each culture was from 4–6 mice). $p$ values are 0.678, 0.001, $4.04 \times 10^{-8}$, and $3.87 \times 10^{-9}$ for R, K$^+$, 3 min and 10 min group, respectively (***$p < 0.001$; **$p < 0.01$; *$p < 0.05$, two-tailed unpaired $t$-test, compared to WT). A schematic showing stimulation and culture fixation time is also shown (lower): EM images were obtained from four conditions in which chemical fixation was applied in the resting condition (R), immediately after 1.5 min application of 90 mM KCl (K$^+$), 3 min or 10 min after KCl application. **c** The number of Ω per bouton before (R) and after KCl application in wildtype (WT) or TM4d-Actb$^{-/-}$ synapses (mean + SEM; each group was from 40–100 synaptic profiles from 3–8 cultures; each culture was from 4–6 mice). $p$ values are 0.298, 0.465, 0.002, and 0.006 (two-tailed unpaired $t$-test, compared to WT) for R, K$^+$, 3 min and 10 min group, respectively. Source data are provided as a Source Data file.

**Dynamin helices surround and constrict Λ's base to mediate Λ→Ω.** After depol$_{1s}$, Λ formed by Flat→Λ and pre-Λ underwent Λ→Ω at a probability [Prob(Λ)$_{→Ω}$] of $0.12 \pm 0.01$ (from 513 cells; Fig. 4a, b)[18]. This transition was due to constriction of ~300–1000 nm Λ-base at a rate of $160 \pm 28$ nm/s (20–80%, $n = 37$; Fig. 4a, b)[18]. The resulting Ω with a visible pore (Ω$_p$, >~60 nm, our STED resolution) sometimes continued to constrict its pore until it became a non-visible pore (Ω$_{np}$, <~60 nm, Fig. 4a)[18]. Although Ω$_{np}$'s pore was not visible, it remained open because it was permeable to A532 (see below for differentiating Ω$_{np}$ from O)[18].

Two sets of evidence suggest that dynamin provides the constriction force to mediate Λ→Ω. First, Prob(Λ)$_{→Ω}$ measured after depol$_{1s}$ was substantially reduced by dynasore (64 cells, Fig. 4b) or overexpressed dynamin 1 K44A (106 cells, Fig. 4b), but not by Latrunculin A or cytochalasin D (Supplementary Fig. 4a, b). Second, before Λ→Ω, dynamin 2-mTFP1 surrounded Λ (Fig. 3d, e); and during Λ→Ω, dynamin 2-mTFP1 was further recruited to Λ/Ω's base/pore region and above (Fig. 4c, 9 out of 9 events; Supplementary Movie S2). Strikingly, the distance between dynamin puncta flanking the base/pore region (d$_{dyn}$) decreased in parallel with Λ/Ω's base/pore size reduction (Fig. 4c, white arrows; Fig. 4d, 9 out of 9 events). The extent of base/pore size reduction was proportional to d$_{dyn}$ reduction (Fig. 4c, d). These results revealed dynamin in the act of constricting large Λ/Ω's base/pore. STED XZ/Y$_{stack}$ imaging confirmed that dynamin 1-mTFP1 (or dynamin 2-mTFP1) surrounded PH$_G$-labeled pre-Ω's pore region (Fig. 4e, f; 65 out of 67 Ω, 17 cells), further supporting our conclusion that dynamin surrounds and thus constricts Λ's base to convert Λ to Ω, and then continues to constrict Ω's pore (Fig. 4g).

To determine whether dynamin alone forms helices surrounding and constricting Λ's base as large as ~800 nm, dynamin 1 was incubated with DOPS (1,2-dioleoyl-sn-glycero-3-phospho-L-serine) liposomes. CryoEM showed that incubation for 1–5 min and 1 hour resulted in dynamin-coated liposomes with a mean of

$97 \pm 6$ nm (range: ~41–301 nm, $n = 104$) and $57 \pm 2$ nm (range: ~40–106 nm, $n = 104$), respectively, which were much smaller than the liposomes without dynamin ($242 \pm 7$ nm, range: ~121–423 nm, $n = 104$; Fig. 5a, b). T-shape structures at the edge of decorated liposomes and the striation pattern seen on liposomes suggest that dynamin assembles on the membrane surface in helical structures (Fig. 5c). We also observed dynamin coating of liposomes as large as ~500–1000 nm under negative staining, in which liposomes were larger (Supplementary Fig. 5).

To observe high-resolution 3-dimensional views of dynamin assembled on vesicles, we generated cryo-electron tomograms (cryoET) of dynamin 1 bound to DOPS liposomes (Fig. 5d; Supplementary Movie S3). CryoET data revealed that dynamin assembles around the large vesicles (~85–166 nm in diameter; Fig. 5d, e; Supplementary Movies S3–S5) that overtime transform into dynamin decorated tubes. Dynamin wraps around vesicles as helices as evident from the T-shaped structures observed in slices of the tomogram that extend into helical ribbons of densities on all sides of vesicles as observed by tomogram segmentation (Fig. 5e; Supplementary Movies S4 and S5). These helices are clearly visible in the area where tube formation starts from the liposomes and also around the side view of the vesicles where dynamin forms hints of helices around the membrane (Fig. 5e; Supplementary Movies S4 and S5). The continuous density of helices around vesicles are not visible due to the tilting limitation of the sample in the microscope but the claw-like structures on both sides of the vesicle clearly suggest a helical pattern (Fig. 5e; Supplementary Movies S4 and S5). We conclude that dynamin assembles as helical arrays around large liposomes that overtime leads to the formation of narrow tubes. This mechanism may underlie dynamin-mediated constriction of Λ's base as large as ~800 nm that converts Λ to Ω in live cells (Fig. 5f).

**Dynamin helices surround and constrict Ω's pore to mediate Ω→O.** After depol$_{1s}$, the pore of pre-Ω and Ω formed by

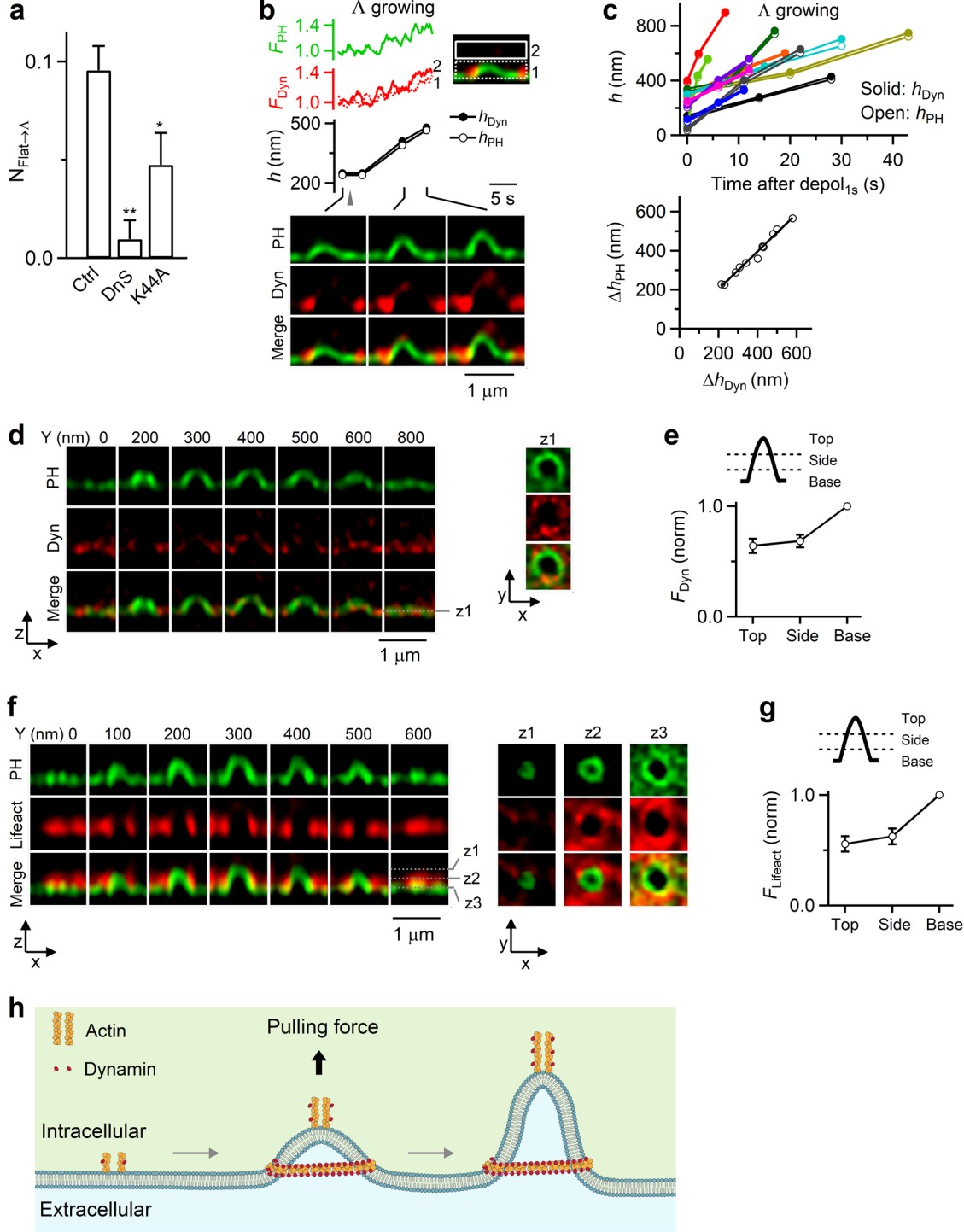

**h**

Actin
Dynamin

Pulling force

Intracellular

Extracellular

Flat→Λ→Ω and pre-Λ→Ω may constrict, resulting in Ω→O transition at a probability [Prob(Ω)→O] of $0.24 \pm 0.02$ (Fig. 6a–e, from 513 cells)[18]. Since Ω was either $\Omega_p$ (pore size >~60 nm) or $\Omega_{np}$ (pore size <~60 nm), Ω→O referred to either $\Omega_p$→O (pore size: 120–375 nm; mean = $212 \pm 30$ nm, 8 $\Omega_p$; Fig. 6a) or $\Omega_{np}$→O (121 $\Omega_{np}$; Fig. 6b)[18]. $\Omega_p$'s pore constriction was sometimes incomplete, resulting in $\Omega_p$→$\Omega_{np}$ (Fig. 6c).

In these transitions, Ω→O was detected as A532 fluorescence ($F_{532}$, strongly excited) dimming, because pore closure prevented bleached A532 contained in the O-shape vesicle from exchanging with fluorescent A532 in the bath. At the same time, the $PH_G$ fluorescence ($F_{PH}$) sustained or decayed with a delay during

A532/$PH_G$ XZ/$Y_{fix}$ scanning (with continuous strong excitation on A532)[18]. Accordingly, $\Omega_p$→$\Omega_{np}$ was detected as pore constriction without $F_{532}$ dimming (Fig. 6c, see also Fig. 4a). During $\Omega_p$→O or $\Omega_p$→$\Omega_{np}$, pore constriction at a rate of $144 \pm 50$ nm/s ($n = 13$) was detected (Fig. 6d).

Four sets of evidence suggest that dynamin mediates Ω→O by constricting Ω's pore. First, $\Omega_p$'s pore constriction rate during $\Omega_p$→O or $\Omega_p$→$\Omega_{np}$ was similar to Λ's base constriction rate during dynamin-dependent Λ→Ω (Fig. 6d), consistent with a similar mechanical force constricting Λ's base and Ω's pore. Second, Prob(Ω)→O (after depol₁ₛ) was substantially reduced by dynasore or dynamin 1 K44A (Fig. 6e), but not by Latrunculin A

**Fig. 3 Dynamin in mediating Flat→Λ in chromaffin cells. a** $N_{Flat→Λ}$ in control (Ctrl, 513 cells from 168 cultures; 336 bovines), in the presence of dynasore (DnS, 80 μM, bath, 20–30 min, 64 cells from 13 cultures; 26 bovines; $p = 0.003$), or with overexpressed dynamin 1-K44A (Dyn-K44A, 106 cells from 21 cultures; 42 bovines; $p = 0.039$). *$p < 0.05$; **$p < 0.01$ (two-tailed unpaired $t$-test, compared to Ctrl). Data are presented as mean + SEM. **b** $F_{PH}$, dynamin 1-mTFP1 fluorescence ($F_{Dyn}$), $PH_G$-labeled Λ's $h_{PH}$, the highest point of dynamin 1-mTFP1 fluorescence associated with Λ ($h_{Dyn}$), and sample $XZ/Y_{fix}$ images showing recruitment of dynamin while Λ is growing. $F_{Dyn}$ from region 1 and 2 (inset) are plotted. $h_{PH}$ and $h_{Dyn}$ were measured as the height from the $PH_G$-labeled base membrane. **c** Upper: $h_{PH}$ and $h_{Dyn}$ (h) plotted versus time after $depol_{1s}$ when Λ was growing. Each color represents one Λ growing event (11 events). Lower: the increase of $h_{PH}$ ($Δh_{PH}$) plotted versus the increase of $h_{Dyn}$ ($Δh_{Dyn}$, left) for each Λ growing event (11 events). The line is a linear regression fit. **d** Dynamin surrounds Λ: XZ images of $PH_G$ and Dyn (dynamin 1-mTFP1) along Y-axis every 100–200 nm as labeled; XY images at a Z-axis location ($z_1$, dotted line) are also shown (right). **e** $F_{Dyn}$ at Λ's base, side and top region (see drawings; 24 Λ; from 17 cells, 4 cultures, 8 bovines). $F_{Dyn}$ is normalized (norm) to that at the base. Data are presented as mean ± SEM. **f** F-actin surrounds Λ: XZ images of $PH_G$/lifeact-mTFP1 (green, red and merge) along Y-axis every 100 nm; XY images at 3 Z-axis locations ($z_1$, $z_2$, $z_3$, and dotted lines) are also shown (right). **g** Lifeact-mTFP1 fluorescence ($F_{lifeact}$) at the base/pore, side, and top region (see drawings) of Λ (18 Λ; from 13 cells, 3 cultures, 6 bovines). $F_{lifeact}$ is normalized (norm) to that at the base/pore region. Data are presented as mean ± SEM. **h** Schematic drawing: actin and dynamin mediate Flat→Λ by generating a pulling force at the center and the growing Λ's tip, and by surrounding Λ's base. Source data are provided as a Source Data file.

or cytochalasin D (Supplementary Fig. 4b). Third, dynamin was present before pore constriction and further recruited to the pore region as $Ω_p$'s pore was constricted (Fig. 6f, see also Fig. 3c, 10 out of 10 events). The distance between dynamin puncta flanking the pore region decreased in parallel with the pore size reduction, revealing dynamin in the act of constricting large $Ω_p$'s pores (Figs. 4c, 6f, g; Supplementary Movie S2). We also observed dynamin puncta at the pore region during $Ω_{np}→O$ (10 out of 11 events), which may disappear sometimes afterwards. $XZ/Y_{stack}$ imaging showed that dynamin fluorescence surrounded $Ω$'s pore (Fig. 4e, f, 65 out of 67 Ω, 17 cells). Thus, dynamin is at the pore region to mediate pore constriction and closure. Fourth, cryoEM showed that dynamin formed helices surrounding and constricting large liposomes and converted them into narrow tubes (Fig. 5). We conclude that dynamin forms helices surrounding and constricting Ω's pore that can be hundreds of nanometers, mediating $Ω→O$ in live cells (Fig. 6h).

**Modeling of flat-to-Ω transition mediated by point-pulling and base constriction.** Our results suggest that at the first stage of budding, dynamin and F-actin pull at the tip of growing Λ−shape and mediate Flat→Λ transition while the Λ's base is confined from growing (Fig. 3h). Subsequently, dynamin constricts Λ's base to mediate Λ→Ω transition (Fig. 4g), which is followed by the closure of Ω's pore and generation of a vesicle (Fig. 6h). In other words, two forces with different directions, a pulling force, and a base/pore constricting force, underlie the flat-to-round transformation. To determine whether these two forces are indeed sufficient to transform a flat membrane into Λ- and Ω-shapes, we computationally analyzed the sequence of membrane shape transformations driven by these forces in the absence of other curvature-generating factors such as a protein coat. Our computational model synthesized elements of previous models aimed at theoretical analysis of endocytic bud formation and scission upon various conditions in yeast and mammalian cells[31–36]. The specificity of our model is in an explicit accounting for the vesicle formation with constriction of the bud base by a protein structure, and a point force at the bud tip in the absence of protein coat on the bud surface. We considered an initially flat circular membrane fragment, referred below to as the endo-site. The endo-site membrane is characterized by a bending rigidity, κ, and is subject to a lateral tension, γ. The endo-site center is lifted by a point pulling force, $f_{pull}$, whereas the edge of the endo-site base is subject to a constraint setting the base radius, $r_b$ (Fig. 7a and Supplementary Fig. 6a–c). The endo-site equilibrium shapes, including Flat, Λ-shape and Ω-shapes (Fig. 7b), were found through minimization of the membrane elastic energy using the theory of membrane bending elasticity (see Methods). The equilibrium shape (Fig. 7b) depended on: (i) the endo-site height,

$H$, which is controlled by $f_{pull}$ (Fig. 7c); (ii) the base radius, $r_b$, and (iii) the intrinsic length-scale of the system, $\left( r_i = \sqrt{\frac{κ}{2γ}} \right)$, that sets the decay length of the membrane bending deformations.

Modeling showed that an increasing $f_{pull}$ applied to the endo-site center upon a constant $r_b$ generates Flat→Λ transition (Fig. 7c, Supplementary Movie S6), whereas subsequent constriction of the normalized base radius, $r_b/r_i$, upon nearly constant $f_{pull}$ mediates Λ→Ω transition and reduces Ω's pore size (Fig. 7d, Supplementary Movie S6). The model's assumptions that $r_b$ is constant during Flat→Λ, but reduces during Λ→Ω are based on live-cell observations that Λ grows without significant base changes (Fig. 1d) and that Λ-base constriction mediates Λ→Ω (e.g., Fig. 4a) (for detail, see ref. [18]). Supplementary Movie S6 shows a simulated continuous Flat→Λ→Ω transition driven, as mentioned above, by just two factors: the pulling force, $f_{pull}$, and a constraint imposed on the endo-site base that reduces its radius, $r_b$.

To verify that these theoretical modeling results apply to live cells, we performed fitting of the computed shape profiles to the experimentally observed ones during Flat→Λ (Fig. 7e, Supplementary Fig. 7a, b, Supplementary Movie S7) and Λ→Ω transition (Fig. 7f, Supplementary Figs. 7a, c, Supplementary Movie S8, see Methods). An excellent agreement between the computational and observed profiles was obtained for the pulling force $f_{pull}$ increasing up to ~3 pN during Flat→Λ transition (Fig. 7e, Supplementary Movie S7), the base radius $r_b$ decreasing during Λ→Ω transition (Fig. 7f, Supplementary Movie S8), and the tension γ being ~0.5–1.2 μN/m. The just few piconewton value of the pulling force suggests that the Flat→Λ transition is mediated by a small number of force-generating molecules associated with the actin-dynamin filaments. Taken together, theoretical modeling (Fig. 7a–d), and quantitative comparison of the model predictions with live-cell data (Fig. 7e, f) support the mechanism in which a point-pulling force mediates the Flat→Λ transitions, whereas a base-constricting force mediates the Λ→Ω transition.

## Discussion

The present work revealed crucial mechanical roles of F-actin and dynamin in mediating Flat→Λ→Ω→O membrane shape transformation in the course of endocytic budding. For Flat→Λ transition, we showed that 1) inhibition of F-actin suppressed Flat→Λ transition in chromaffin cells where Λ grew within a boundary-confined base (Fig. 1), 2) knockout of β-actin inhibited Λ formation at hippocampal boutons (Fig. 2), and 3) inhibition of dynamin also inhibited Flat→Λ in chromaffin cells (Fig. 3). We caught F-actin and dynamin in generating Λ: they were recruited

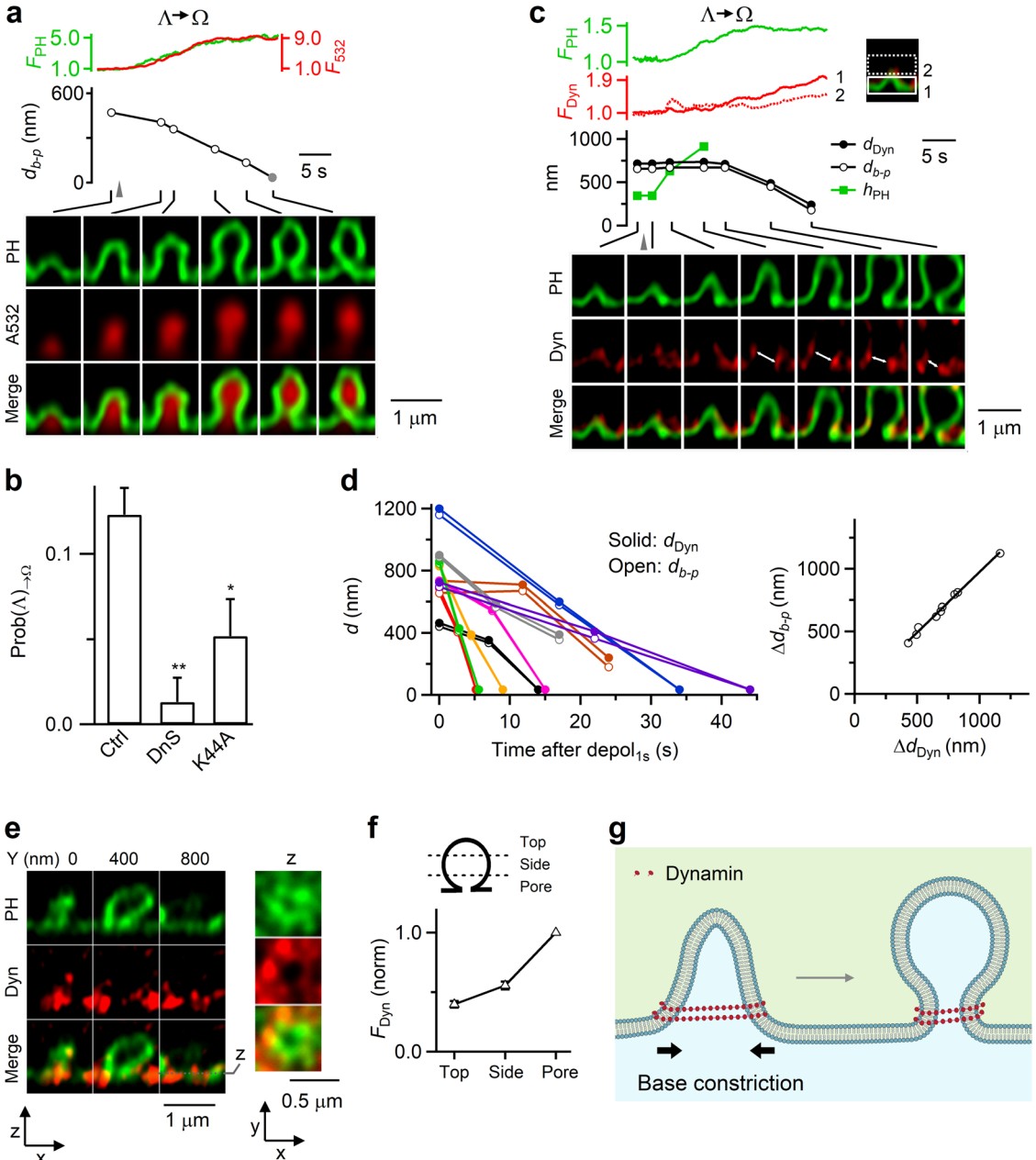

**Fig. 4 Dynamin mediates Λ→Ω by surrounding and constricting Λ's base in chromaffin cells. a** $F_{PH}$, $F_{532}$, Λ's base distance or Ω's pore diameter ($d_{b-p}$), and XZ/$Y_{fix}$ images (at times indicated with lines) showing $d_{b-p}$ constriction during Λ→Ω. Gray triangle, $depol_{1s}$. Gray circle: <60 nm (STED resolution). **b** Inhibition of dynamin reduces Λ→Ω transition: Prob(Λ)$_{→Ω}$ in control (Ctrl, 513 cells from 168 cultures; 336 bovines), in the presence of dynasore (DnS, 64 cells from 13 cultures; 26 bovines; $p = 0.007$), overexpressed dynamin 1-K44A (Dyn-K44A, 106 cells from 21 cultures; 42 bovines; $p = 0.032$). *$p < 0.05$; **$p < 0.01$ (two-tailed unpaired $t$-test, compared to Ctrl). Data are presented as mean + SEM. **c** Dynamin puncta flanked and moved with constricting Λ's base and constricting Ω's pore: $F_{PH}$, $F_{Dyn}$, Λ's base distance or Ω's pore diameter ($d_{b-p}$), the distance between two dynamin puncta flanking Λ/Ω's base/pore ($d_{Dyn}$, double arrows), and sample XZ/$Y_{fix}$ images of PH/dynamin for a Λ→Ω transition. $F_{Dyn}$ (normalized) from region 1 (near Λ/Ω base) and 2 (above base, inset) are plotted. **d** Left: $d_{b-p}$ and $d_{Dyn}$ ($d$) plotted versus time after $depol_{1s}$ during Λ→Ω transition. Each color: one event (9 events). Right: the decrease of $d_{b-p}$ ($\Delta d_{b-p}$) plotted versus the decrease of $d_{Dyn}$ ($\Delta d_{Dyn}$) observed during Λ→Ω transition (each circle: one transition, 9 transitions). The line is a linear regression fit. **e** Dynamin surrounds Ω's pore region: XZ images of $PH_G$ and Dyn (dynamin 1-mTFP1) for a $Ω_p$ along Y-axis every 400 nm as labeled; XY images at the Z-axis location, z (dotted line), are also shown. **f** $F_{Dyn}$ at the pore, side and top region of Ω (see drawings, 67 Ω; from 17 cells, 4 cultures, 8 bovines). $F_{Dyn}$ is normalized (norm) to that at the pore region. **g** Schematic drawing: dynamin surrounds and constricts Λ's base, converting Λ to Ω. Source data are provided as a Source Data file.

to surround the growing Λ's base, side, and tip, formed filaments attached at growing Λ's tip and at the spike-like protrusions from Λ's tip, and moved inward in parallel with Λ's tip and spike-like protrusion (Figs. 1 and 3). These results suggest that F-actin filament and dynamin mediate Flat→Λ by pulling membrane

inward at least in part at the growing Λ's tip to generate spike-like membrane protrusions while surrounding the growing Λ's base to prevent Λ's base from growing. Realistic quantitative modeling further supported this conclusion by showing that a point-pulling force of ~3 pN at the center of a confined endocytic zone can in

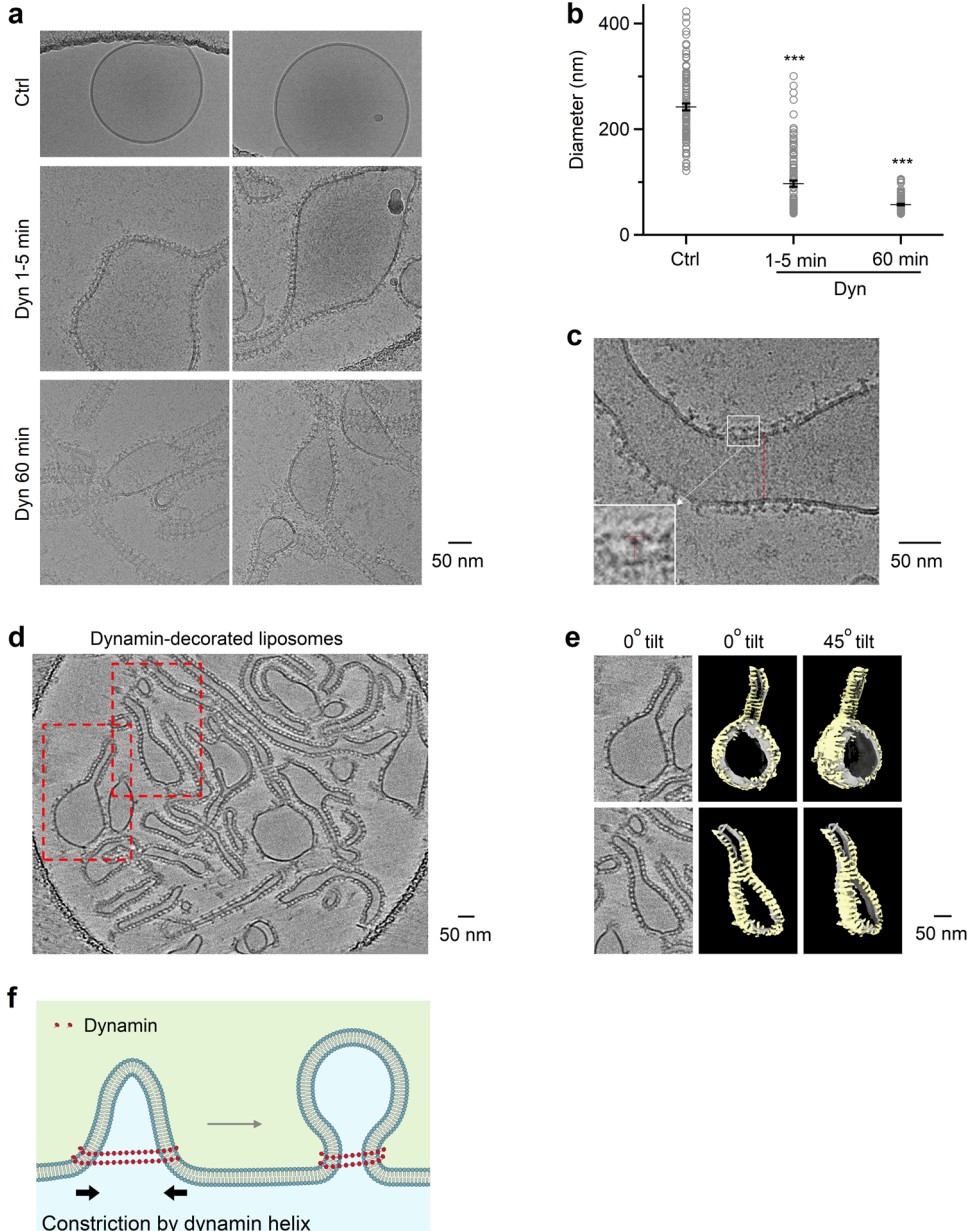

**Fig. 5 Dynamin helix reduces liposome size in vitro. a, b** Sampled cryoEM images (**a**) and diameter (**b**) mean ± SEM) of DOPS vesicles coated without (Ctrl, $n = 104$) or with dynamin 1 after dynamin 1 incubation for 1-5 min (Dyn 1-5 min, $n = 104$, $p = 6.75 \times 10^{-39}$) or 60 min (Dyn 60 min, $n = 104$, $p = 5.82 * 10^{-70}$). ***$p < 0.001$, two-tailed unpaired $t$-test (compared to Ctrl); each circle represents a liposome (104 micrographs from 3 independent experiments). **c** Dynamin 1 assembled around large liposomes. The regular pattern of dynamin as 'T' structures on the edge of the lipid bilayer (insert) and the striation pattern seen on the lipid (dotted red line) suggests dynamin is wrapping around the lipid as a helix. **d** Cryo-electron tomogram of dynamin 1 with DOPS vesicles. Several dynamin-decorated vesicles and tubes are apparent with the characteristic T-shaped structure of dynamin extending from the membrane. Red-dashed boxes indicated two regions of the tomogram that were segmented as shown in panel (**e**). **e** CryoET segmentation of two DOPS vesicles with diameters of ~166 nm (top) and ~85 nm (bottom). Left panels, central slice of tomograms with dynamin T-shaped structures around vesicles and tubes. Middle and right panels show 3D segmentation of vesicles at 0°C and ~45°C tilts respectively. As you rotate the segmented volumes, striations of the dynamin density (yellow) are observed indicating a helical assembly of dynamin around the lipid bilayer (grey). **f** Schematic drawing: dynamin helix surrounds and constricts Λ's base, converting Λ to Ω. This drawing is similar to Fig. 4g, except that the label "Base constriction" is changed to "Constriction by dynamin helix", according to results shown in Fig. 5**a–e**. Source data are provided as a Source Data file.

principle mediate Flat→Λ (Fig. 7). For Λ→Ω→O, which is mediated by sequential constriction of Λ's base and then Ω's pore[18], we found that inhibition of dynamin significantly inhibited Λ→Ω and Ω→O (Figs. 4 and 6). We visualized dynamin in constricting Λ's base and Ω's pore: dynamin surrounded and constricted Λ's base to mediate Λ→Ω, and then surrounded and constricted Ω's pore to mediate Ω→O (Figs. 4 and 6). Λ's base

and Ω's pore could be as large as ~630–800 nm. To support the proposal that dynamin alone could mediate constriction of such large Λ/Ω's base/pores, we demonstrated that dynamin forms helices surrounding and constricting liposomes from hundreds to tens of nano-meters (Fig. 5). Realistic quantitative modeling further supports this conclusion by showing that in principle a constriction force that reduces Λ's base and Ω's pore can drive Λ-

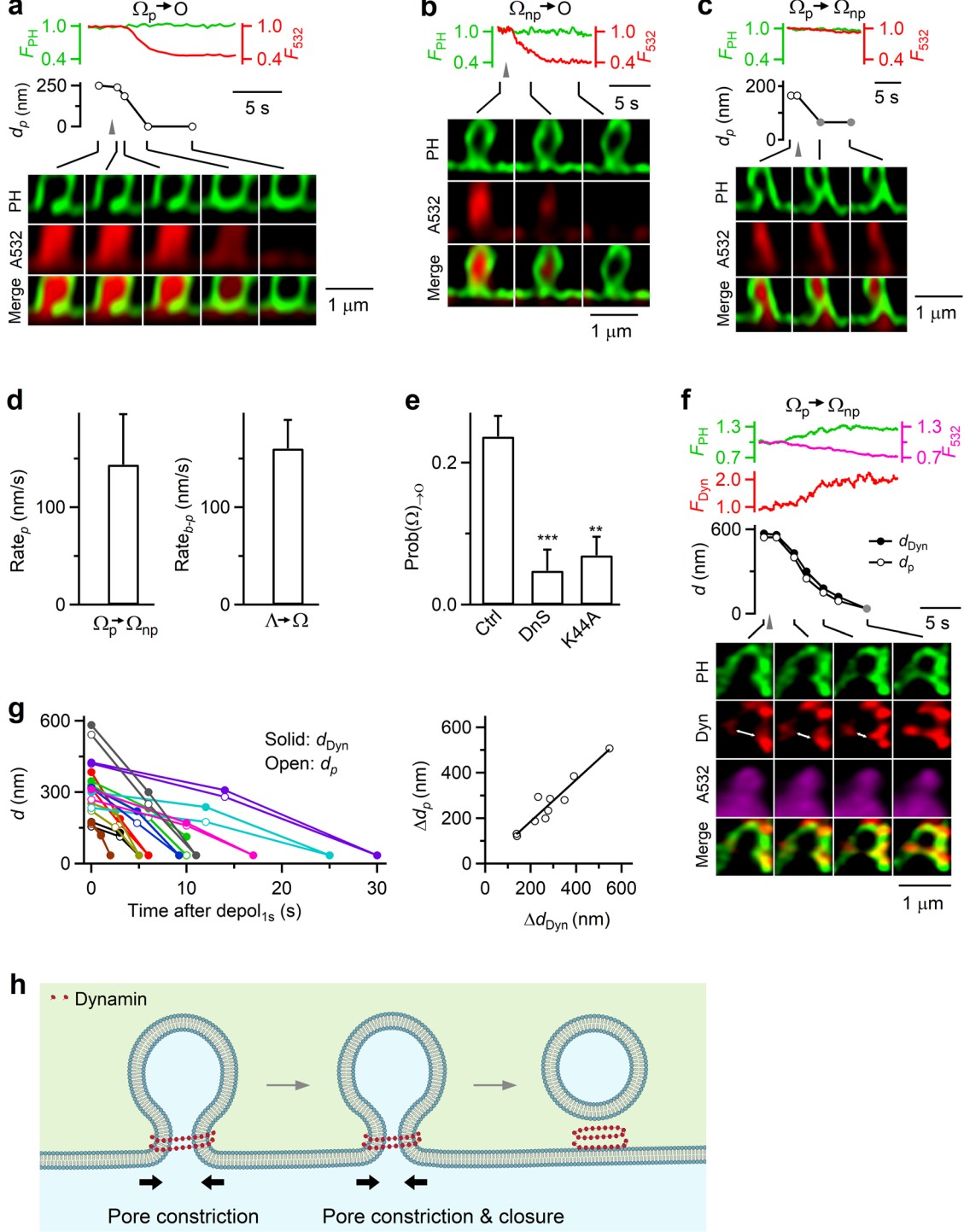

**Fig. 6 Dynamin mediates Ω→O by surrounding and constricting Ω's pore. a** $F_{PH}$, $F_{532}$, Ω's pore diameter ($d_p$), and XZ/Y$_{fix}$ images showing $Ω_p$→O. Gray triangle: depol$_{1s}$. **b** $F_{PH}$, $F_{532}$, and XZ/Y$_{fix}$ images showing $Ω_{np}$→O. **c** $F_{PH}$, $F_{532}$, Ω's pore diameter ($d_p$), and XZ/Y$_{fix}$ images showing Ω's pore constriction from visible to non-visible pore ($Ω_p$→$Ω_{np}$). Gray triangle: depol$_{1s}$. gray circle: <60 nm. **d** The rate of pore size decrease during $Ω_p$→$Ω_{np}$ (Rate$_p$, 20–80%, $n = 13$) and the rate of $d_{b-p}$ decrease during Λ→Ω (Rate$_{b-p}$, 20–80%, $n = 22$) are similar. Data are presented as mean + SEM. **e** Inhibition of dynamin reduces Ω→O transition: Prob(Ω)$_{→O}$ in control (Ctrl, 513 cells from 168 cultures; 336 bovines), in the presence of dynasore (DnS, 64 cells from 13 cultures; 26 bovines; $p = 0.0009$) or with overexpressed dynamin 1-K44A (Dyn-K44A, 106 cells from 21 cultures; 42 bovines; $p = 0.0018$). **$p < 0.01$; ***$p < 0.001$ (two-tailed unpaired $t$-test, compared to Ctrl). Data are presented as mean + SEM. **f** Dynamin puncta flanked and moved with the constricting pore during $Ω_p$→$Ω_{np}$: $F_{PH}$, $F_{Dyn}$, $Ω_p$'s pore size ($d_p$), the distance between two dynamin puncta flanking the pore ($d_{Dyn}$, double arrows), and sampled XZ/Y$_{fix}$ images of PH/dynamin. Gray circles: <60 nm (STED resolution). **g** Left: Ω's $d_p$ and $d_{Dyn}$ plotted versus time after depol$_{1s}$ during $Ω_p$→$Ω_{np}$. Each color: one $Ω_p$→$Ω_{np}$ transition. Right: reduction of Ω's $d_p$ ($Δd_p$) plotted versus reduction of $d_{Dyn}$ ($Δd_{Dyn}$) for each $Ω_p$→$Ω_{np}$ transition (each circle: one transition). **h** Schematic drawing: dynamin mediates Ω→O by surrounding and constricting Ω's pore. Source data are provided as a Source Data file.

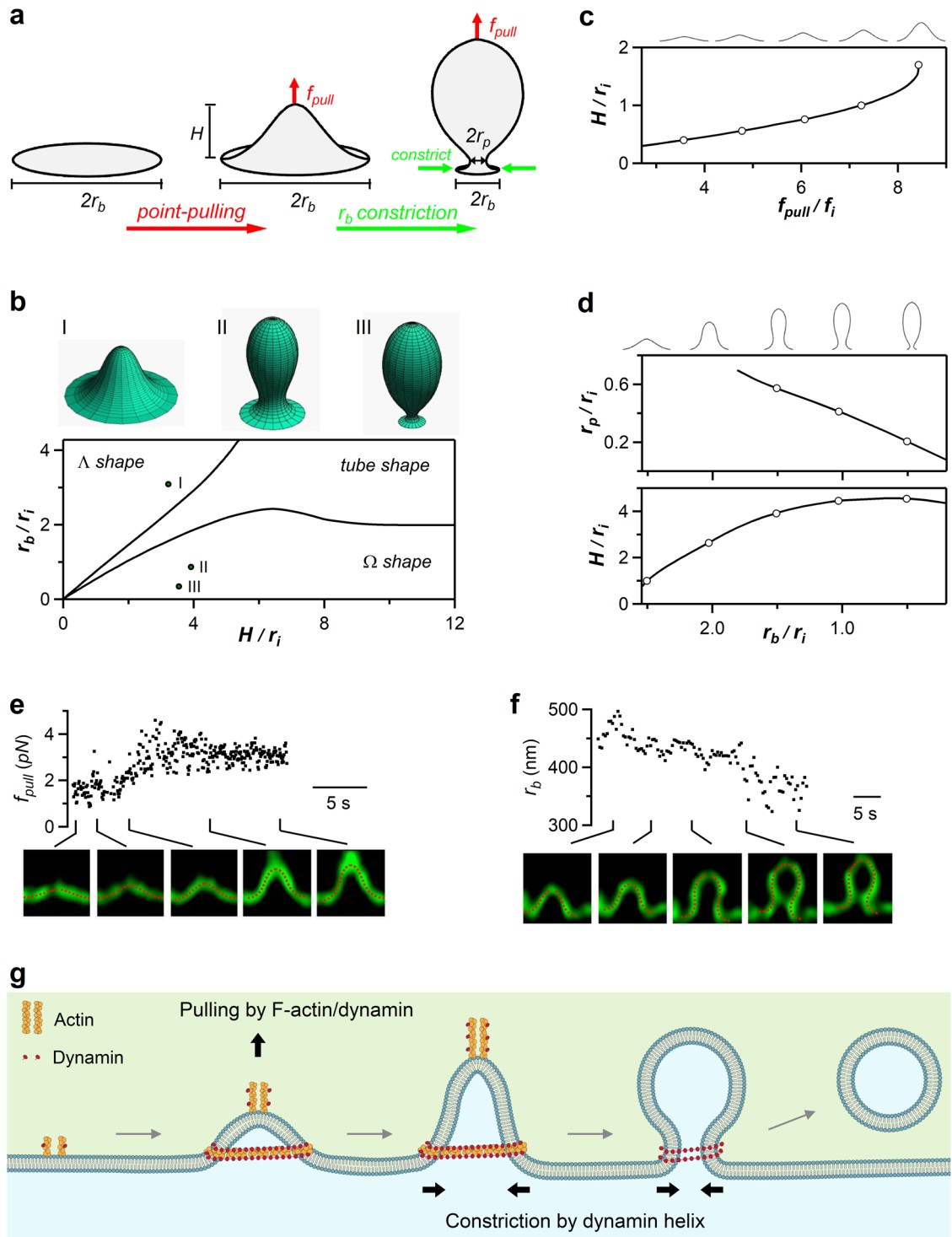

to-Ω transition and constrict Ω's pore (Fig. 7). We concluded that 1) the actin filaments together with dynamin produce pulling force to mediate Flat→Λ transition, 2) the dynamin helices surround and constrict Λ's base as large as hundreds of nano-meters in diameter to mediate Λ→Ω transition, and 3) the dynamin helices continue to constrict Ω's pore as large as hundreds of nano-meters in diameter to mediate Ω→O transition (Fig. 7g).

Our results suggest that dynamin is not just a fission enzyme cutting narrow (~5–20 nm) necks as previously thought[20], but a master player orchestrating every modular transition of the flat-to-round transformation, including Flat→Λ, Λ→Ω and Ω→O. Dynamin and F-actin co-mediate Flat→Λ by pulling the

membrane inward, which raises a question about the mode of the cooperation between the two proteins. Dynamin has been shown to interact with F-actin at the tail of actin comets, podosomes, filopodia, and in F-actin bundles regulating the dynamics and mechanical strength of the actin cytoskeleton during cell fusion[20,28–30]. With these previous findings and our observation that dynamin and F-actin were localized at and above the growing Λ's tip and its spike-like protrusions (Figs. 1 and 3), we proposed that the role of dynamin is to mediate and enhance the bundling of F-actin filaments at the tip of the growing Λ-shapes (Fig. 3h).

Our finding that dynamin converts Λ into Ω by constricting the ~800 nm large Λ-shape base is rather surprising (Fig. 4), given

**Fig. 7 Modeling of Flat→Λ→Ω shape transition mediated by a centre-pulling force and a base-constriction force. a** Drawing for modeling two forces underlying the membrane shape transformations: a point-pulling force ($f_{pull}$) converts the flat membrane to Λ-shape, whereas a constriction force reduces the Λ-shape base radius ($r_b$), which transforms Λ- to Ω-shape. $H$ is the height; $r_p$ is the pore radius. **b** Lower: computed shape diagram – membrane shape (Λ-, Ω- or tube-shape) depends on $H$ and $r_b$, both normalized to the internal length scale $r_i(=\sqrt{\frac{\kappa}{2\gamma}})$. Upper: examples of computed Λ shape I, ($H = 3.0 \cdot r_i$, $r_b = 3 \cdot r_i$), Ω shape with a large pore II, $H = 3.9 \cdot r_i$, $r_b = 0.81 \cdot r_i$, and Ω shape with a narrow pore III, $H = 3.8 \cdot r_i$, $r_b = 0.46 \cdot r_i$. **c** Computed Flat→Λ transition. An increase of $f_{pull}$ raises Λ-shape height $H$ (normalized to $r_i(=\sqrt{\frac{\kappa}{2\gamma_0}})$) upon a constant base radius $r_b(= 2.5 \cdot r_i)$. $f_{pull}$ is normalized to the intrinsic force $f_i = \sqrt{2\kappa\gamma_0}$. Circles represent Λ-profiles shown on the top (applies to panels c and d). **d** Computed Λ→Ω transition. Constriction of the base radius, $r_b$ (normalized to $r_i(=\sqrt{\frac{\kappa}{2\gamma_0}})$), upon keeping the force ($f_{pull} = 7.2 \cdot f_i$) and membrane area constant converts Λ to Ω shape and reduces the radius of the Ω-shape pore, $r_p$ (normalized to $r_i$). **e** Upper: $f_{pull}$ increased during an experimentally observed Flat→Λ-shape transition (each point from a frame in Supplementary Movie S7). Lower: sampled Λ-shape observed (green) and computed (red) that matches the observed one. $f_{pull}$ (upper) was obtained from computed Λ-shape (see Methods). Sticks indicate the time at which the sampled images were taken (also applies to panel F). **f** Upper: the endo-site base radius $r_b$ decreased in the course of an experimentally observed Λ→Ω endo-site base radius transition (each point from a frame in Supplementary Movie S8). Lower: examples of the observed Λ and Ω (green) and their computational fittings (red). $r_b$ (upper) was obtained from the fitting. **g** Schematic summary: actin filaments and dynamin generate a pulling force to mediate Flat→Λ transition; dynamin helix forms rings to surround and constrict Λ's base and Ω's pore, mediating Λ→Ω→O shape transition. Source data are provided as a Source Data file.

that dynamin has been considered to surround and drive fission of narrow (~5–20 nm) membrane necks[20]. However, our finding is further supported by the observation that dynamin can constrict Ω's pore from diameters as large as hundreds of nanometers to closure (Fig. 6), that dynamin alone can form helices surrounding and constricting liposomes from hundreds to tens of nano-meters (Fig. 5), and that dynamin can constrict fusion pores from diameters as large as ~490 nm to closure[24]. Does dynamin constrict large Λ/Ω's base/pore in the same way as it constricts narrow pores? Structural studies suggest that conformational changes of dynamin's α-helical bundle domains generate constriction of the narrow neck underneath dynamin helices[20,21]. Evidently, one such conformational change is insufficient to account for the constriction of Λ/Ω's base/pore that can be, initially, hundreds of nano-meters wide. Given that large pore constriction can take place within seconds or milliseconds (Fig. 6)[18,24], multiple and rapid repeats of such conformational changes are needed to constrict the wide bases and pores of Λ and Ω. Alternatively, mechanisms other than narrow-pore constriction may underlie the large base/pore constriction. The way in which dynamin constricts large base/pore would be a fascinating research topic for the future.

In mammalian cells, actin is suggested to facilitate endocytosis by overcoming the plasma membrane tension[37]. Although F-actin has been implicated in pit formation during clathrin-independent endocytosis[5–7], the exact physical force F-actin provides is unclear due to difficulty of real time dissecting the intermediate structures and the type of physical force(s) required to form a Ω-shape pit. With real-time observation of Flat→Λ→Ω, we found that F-actin filaments provide a pulling force for Λ formation applied to the tip of the growing Λ. This finding not only provides real-time imaging data supporting actin involvement in pit formation[5–7], but also may explain why and how actin is involved in many endocytic modes in mammalian cells, such as receptor-mediated endocytosis, ultrafast, fast or slow endocytosis, and bulk endocytosis[14,16,27,37,38].

The role of actin proposed here is different from that suggested for endocytosis in cell-wall-containing yeast, wherein actin polymerization from the plasma membrane towards the cytosol is thought to generate pushing forces, which are distributed along the surface of a clathrin cage and serve to elongate the bud neck[2,39,40]. Our suggestion is supported by the real-time observation of F-actin recruitment, attachment and movement with the growing Λ's tip and the spike-like protrusion (Fig. 1). Since we also observed F-actin recruitment at the growing Λ's base and side (Figs. 1 and 3), it is possible that actin-polymerization-generated pushing force also contributes to mediating Flat→Λ in mammalian cells. For yeast endocytosis, since actin filaments

were also found at the tip of the pit[2,39,40], F-actin filaments may also generate pulling forces at the tip to elongate the pit.

Our findings raise the questions as to how F-actin filaments generate a pulling force. One possibility is that it is generated by a network of actin filaments that surround the membrane bud and undergoes polymerization at the bud base[32–34,36]. The force resulting from polymerization could be transmitted to the bud apex[33]. Alternatively, the pulling force can be generated by a bundle of contractile complexes of actin and myosin filaments[41], the stress-fibre[42], whose one end is attached to the bud tip, whereas the second end is anchored in the cytoplasmic actin network. Dynamin could mediate the actin filament bundling within the stress fibre[30,43]. Given the relatively large size of the endocytic buds studied here, and the observation that actin is attached to Λ's tip and the associated spike-like protrusions, we favor the latter mode, i.e., the point force generation.

The synergy between F-actin and dynamin in endocytic membrane invagination (Flat→Λ) is in contrast to a recent finding that F-actin facilitates fusion pore expansion whereas dynamin constricts fusion pore and thus counteracts pore expansion in chromaffin cells[24]. This apparent difference might originate from a qualitative difference between the machineries mediating the actin action in these two processes. Fusion pore expansion is facilitated by an enhanced membrane tension, which can be produced and/or modulated by a force pushing the plasma membrane and resulting from polymerization of cortical actin filaments against the membrane[24,44]. In contrast, endocytic membrane invagination may be driven by an actin/dynamin-dependent pulling force applied at Λ's tip, as suggested in the present work.

Dynamin/actin-mediated Flat→Λ→Ω→O transition reported here may account for many clathrin-independent, but dynamin/actin-dependent modes of endocytosis, which mediates cellular uptake of extracellular ligands, receptors, viruses, bacteria, prions, bacterial toxins, and secretory vesicles[1,5]. For example, dynamin and F-actin are involved in ultrafast, fast, slow, bulk, and over-shoot endocytosis that maintain exocytosis and synaptic transmission at synapses and non-neuronal secretory cells[10,14,16,22,27,38,45,46]. These endocytic forms are considered mostly clathrin-dispensable[8–12]. Dynamin/actin-mediated Flat→Λ→Ω→O transition may account for these forms of endocytosis. Our finding that β-actin knockout inhibits Λ formation at hippocampal synapses (Fig. 2) further supports this suggestion.

Our finding of dynamin/actin's role in mediating Flat→Λ→Ω→O transition challenges the current view on clathrin-dependent endocytosis, where clathrin is considered to be the primary factor for pit formation, dynamin is supposed to cut the pit's narrow neck, and actin is suggested to facilitate endocytosis

by overcoming membrane tension[2,3]. We suggest that dynamin and actin participate in generating pulling and constriction forces required for pit formation, whereas membrane coating with clathrin and its partner proteins may confine the pit size to ~40-100 nm, the size of the clathrin-formed cages. Supporting this suggestion, initiation and maturation of fluorescent clathrin spots, which presumably involve pit formation, depend on dynamin and actin, suggesting that dynamin and actin may regulate clathrin-dependent endocytosis before fission[2,3].

Dynamin knockout leads to the accumulation of $\Omega$-profiles and long tubes at the plasma membrane[46–48]. We consider two reasons that may reconcile this EM observation with our live-cell observation that dynamin is involved in not only $\Omega{\to}O$, but also Flat$\to\Lambda$ and $\Lambda{\to}\Omega$. First, the EM observation was obtained from cells lacking dynamin for days to weeks[46–48], whereas Flat$\to\Lambda$, $\Lambda{\to}\Omega$ and $\Omega{\to}O$ transitions were measured within 60 s after depol$_{1s}$ in live cells (Figs. 3, 4, and 6). If dynamin inhibition prolongs Flat$\to\Lambda$ and $\Lambda{\to}\Omega$ transition time from seconds to minutes or hours, but not days or weeks, it may explain why dynamin inhibition reduced Flat$\to\Lambda$ and $\Lambda{\to}\Omega$ transitions measured within 60 s (Figs. 3, 4, and 6), but increased the number of invaginations observed with EM in days or weeks afterwards[46–48]. Similarly, if dynamin inhibition reduces Flat$\to\Lambda$ and $\Lambda{\to}\Omega$ transition by a lesser extent than $\Omega{\to}O$ transition measured in days or weeks after the transition is initiated, accumulation of $\Omega$-profiles by dynamin inhibition would be observed in days or weeks later. Second, $\Lambda$-profiles have not been quantified in cells lacking dynamin[46–48], likely because without real-time observation, it is difficult to determine whether a $\Lambda$-profile stems from an endocytic Flat$\to\Lambda$ transition or from a non-endocytic membrane folding. Furthermore, without real-time observation, a change in the $\Lambda$-profile number reflects the net change of both Flat$\to\Lambda$ and $\Lambda{\to}\Omega$, making it difficult to pinpoint which specific transition is affected. Accordingly, it remains unclear whether dynamin knockout affects $\Lambda$-profile formation or not. Our finding of dynamin involvement in Flat$\to\Lambda$, $\Lambda{\to}\Omega$ and $\Omega{\to}O$ is thus not in conflict with the observation of accumulated invaginations in dynamin knockout cells.

Prob$(\Lambda)_{\to\Omega}$ and Prob$(\Omega)_{\to O}$ are low, ~0.12 and 0.24, respectively (Figs. 4 and 6), leaving many newly generated $\Lambda$ and $\Omega$ to stay at the plasma membrane (without reversing back to flat membrane)[18]. Upon next depolarization, the low Prob$(\Lambda)_{\to\Omega}$ makes preformed-$\Omega{\to}O$ the main transition for vesicle reformation, which, together with kiss-and-run, mediates diverse endocytic modes, including ultrafast, fast, slow, overshoot, and bulk endocytosis[18]. The low Prob$(\Lambda)_{\to\Omega}$ and Prob$(\Omega)_{\to O}$ might be due to the difficulty of significant shape changes mediated by dynamin oligomerization that may surround and constrict large $\Lambda$'s base and $\Omega$'s pore. Considering that $\Lambda$'s base are larger than $\Omega$'s pore, $\Lambda$'s base constriction might take more energy than $\Omega$'s pore constriction, which might explain why Prob$(\Lambda)_{\to\Omega}$ (~0.12) is lower than Prob$(\Omega)_{\to O}$ (~0.24).

In summary, we established a molecular mechanical model for the flat-to-round transformation of membrane shape in the course of endocytic budding, where dynamin and actin provide the center-pulling and base/pore-constriction forces to mediate the sequential Flat$\to\Lambda{\to}\Omega{\to}O$ shape transition (Fig. 7g). This mechanism may govern the endocytosis of secretory cells that maintain exocytosis and synaptic transmission. The impact of this model is well beyond secretory cells, as its principle may be applicable practically to all flat-to-round membrane transformations, such as oval/round organelle formation, vesicle formation from endoplasmic reticulum, Golgi, endosome or lysosome, and multiple membrane processes such as membrane protein/lipid trafficking and viral entry[3,13,49]. The involvement of actin and dynamin in the above-mentioned cellular processes further supports our suggestion that dynamin/actin-mediated Flat$\to\Lambda{\to}\Omega{\to}O$ shape transition is of wide applications to many cell biological processes involving the membrane curvature transition. The powerful mechanical roles of F-actin and dynamin revealed in the present may also shed light on the mode by which F-actin- and/or dynamin mediate curvature transitions other than the flat-to-round transformation, such as those involved in cell migration, cell fusion, cell division, neuronal branching, and cell-shape formation[1,4,49].

## Methods

**Animals**. Animal care and use were carried out according to NIH guidelines and were approved by the NIH Animal Care and Use Committee (NINDS ASP 1170-22). Actb$^{LoxP/LoxP}$ mice were obtained by homozygous breeding using standard mouse husbandry procedures as previously described[16]. *CAGGCre-EM$^{TM}$* mice were obtained from Jackson Lab. Its Cre efficiency induced by tamoxifen reaches 90% in cultured cells (Hayashi and McMahon, 2002). Cre-ER$^{TM}$; Actb$^{LoxP/LoxP}$ mice were generated by crossing *CAGGCre-EM$^{TM}$* mice with Actb$^{LoxP/LoxP}$ mice in our lab[16]. Wild-type (C57BL/6J) and Actb$^{LoxP/LoxP}$ mice at P0 of either sex were used. Mouse housing conditions: temperature: 70–74F; humidity: 35–60%; light cycle: 6AM–6PM and dark cycle: 6PM–6AM.

**Primary bovine chromaffin cell culture and transfection**. Bovine adrenal chromaffin cells containing dense-core vesicles are widely used for exo-endocytosis studies[22,50]. To prepare primary bovine adrenal chromaffin cell culture[23,25], we purchased fresh bovine adrenal glands from a local abattoir (J. W. Treuth & Sons Inc., 328 Oella Ave, Catonsville, MD 21228; web site: https://www.jwtreuth.com), which collected the adrenal glands immediately after the animal's death. The use of bovine adrenal glands for the culture of bovine adrenal chromaffin cells did not require an animal use protocol. Fresh adult (21–27 months old) bovine adrenal glands were immersed in pre-chilled Locke's buffer on ice for transportation to the lab. This solution contained: NaCl, 145 mM; KCl, 5.4 mM; Na$_2$HPO4, 2.2 mM; NaH$_2$PO4, 0.9 mM; glucose, 5.6 mM; HEPES, 10 mM (pH 7.3, adjusted with NaOH). Upon arrival at the lab, glands were perfused with Locke's buffer, then infused with Locke's buffer containing collagenase P (1.5 mg/ml, Roche), trypsin inhibitor (0.325 mg/ml, Sigma) and bovine serum albumin (5 mg/ml, Sigma), and incubated at 37 °C for 20 min. The digested medulla was minced in Locke's buffer, and filtered through a 100 µm nylon mesh. The filtrate was centrifuged (48 × g, 5 min), re-suspended in Locke's buffer and re-centrifuged until the supernatant was clear. The final cell pellet was re-suspended in pre-warmed DMEM medium (Gibco) supplemented with 10% fetal bovine serum (Gibco) and plated onto poly-L-lysine (0.005% w/v, Sigma) and laminin (4 µg/ml, Sigma) coated glass coverslips.

Before plating, some cells were transfected by electroporation using Basic Primary Neurons Nucleofector Kit (Lonza), according to the manufacturer's protocol. The cells were incubated at 37 °C with 9% CO$_2$ and used within 5 days.

**Plasmids and fluorescent dyes**. When Atto 532 (A532, Sigma) was included in the bath solution, the dye concentration was 30 µM. The PH-EGFP (phospholipase C delta PH domain attached with EGFP) was obtained from Dr. Tamas Balla. PH-mNeonGreen (PH$_G$) and PH-mTFP1 construct were created by replacing the EGFP tag of PH-EGFP with mNeonGreen (Allele Biotechnology)[51] or mTFP1 (Addgene), respectively. Dynamin 1-K44A-mRFP was purchased from Addgene (#55795). Dynamin 2-mTFP1 (or dynamin 1-mTFP1) and dynamin 2-mNeonGreen construct were created by replacing the EGFP tag of dynamin 2-EGFP (or dynamin 1-EGFP, Addgene) with mTFP1 or mNeoGreen, respectively. Lifeact-mTFP1 construct was created by replacing the TagGFP2 of lifeact-TagGFP2 (Ibidi) with mTFP1.

**Electrophysiology and the stimulation protocol: 1 depol$_{1s}$ per cell**. At room temperature (20–22 °C), whole-cell voltage-clamp and capacitance recordings were performed with an EPC-10 amplifier together with the software lock-in amplifier (PULSE 8.74, HEKA, Lambrecht, Germany)[25,52]. The holding potential was −80 mV. For capacitance measurements, the frequency of the sinusoidal stimulus was 1000–1500 Hz with a peak-to-peak voltage ≤50 mV. The bath solution contained 125 mM NaCl, 10 mM glucose, 10 mM HEPES, 5 mM CaCl$_2$, 1 mM MgCl$_2$, 4.5 mM KCl, 0.001 mM TTX, and 20 mM TEA, pH 7.3 adjusted with NaOH. The pipette (2–4 MΩ) solution contained 130 mM Cs-glutamate, 0.5 mM Cs-EGTA, 12 mM NaCl, 30 mM HEPES, 1 mM MgCl$_2$, 2 mM ATP, and 0.5 mM GTP, pH 7.2 adjusted with CsOH. These solutions pharmacologically isolated calcium currents.

For stimulation, we used a 1-s depolarization from the holding potential of −80 mV to +10 mV (depol$_{1s}$). We used this stimulus because it induces robust exo-endocytosis as reflected in capacitance recordings (Fig. 1a)[25,53,54]. Since prolonged whole-cell recording slows down endocytosis[55], we limited to 1 depol$_{1s}$ per cell.

**STED imaging**. STED images were acquired with Leica TCS SP8 STED 3× microscope that is equipped with a 100 × 1.4 NA HC PL APO CS2 oil immersion objective and operated with the LAS-X imaging software. Excitation was with a tunable white light laser and emission was detected with hybrid (HyD) detectors. $PH_G$ and A532 were sequentially excited at 485 and 540 nm, respectively, with the 592 nm STED depletion beam, and their fluorescence collected at 490–530 nm and 545–587 nm, respectively.

For three-color STED imaging with 592 nm STED depletion laser, dynamin 2-mTFP1 (or dynamin 1-mTFP1 or Lifeact-mTFP1), $PH_G$, and A532 were excited at 442 nm (power: 1–3 mW), 507 nm (power: 1–5 mW), and 545 nm (power: 4–6 mW), respectively, and their fluorescence collected at 447–502 nm, 512–540 nm, and 550–587 nm, respectively. For two-color STED imaging of dynamin 2-mTFP1 (or dynamin 1-mTFP1 or Lifeact-mTFP1) and $PH_G$ with the 592 nm STED depletion beam, mTFP1, and $PH_G$ were sequentially excited at 442 nm (power: 1–3 mW) and 507 nm (power: 1–5 mW), respectively, and their fluorescence collected at 447–505 nm and 516-587, respectively. STED XZ-plane imaging was performed with 3D depletion patten [z(3D)-STED]. The depletion laser power distribution of depletion 3D doughnut was 60% or higher in Z direction and 40% or lower in XY direction. The depletion laser power was 100–500 mW for imaging of $PH_G$, dynamin 2-mTFP1 or lifeact-mTFP1, but 100-200 mW for imaging of A532.

The excitation laser power for A532 was 4–6 mW, at which fluorescent A532 can be bleached within a few seconds during $XZ/Y_{fix}$ imaging every 26–200 ms. This feature was used to distinguish whether the $PH_G$-labeled Ω-profile's pore is closed or not, because pore closure prevents bleached A532 (caused by strong excitation) from exchange with fluorescent A532 in the bath, resulting in A532 spot fluorescence decay[24,25]. In contrast, an open-pore does not cause A532 spot fluorescence decay, because an open-pore allows for continuous exchange of bleached A532 in the Ω-profile with fluorescent A532 in the bath[24,25].

We labeled membrane with PH-mNeonGreen (PH-mNG, excited at 511 nm, fluorescence collected at 516–587 nm), but labeled F-actin and dynamin with lifeact-mTFP1 and dynamin 2-mTFP1 (excited at 442 nm, fluorescence collected at 447–505 nm). If fluorescence resonance energy transfer (FRET) between mTFP1 and mNG takes place, the admitted fluorescence of mTFP1 is absorbed by nearby mNeonGreen within ~10 nm. Lifeact-mTFP1 and dynamin 2-mTFP1 fluorescence could be underestimated, but the observed mTFP1 fluorescence should reflect their minimal distribution. Hence, our conclusion that F-actin and dynamin are physically available near the membrane to mediate pulling and constriction should not be affected by the potential FRET between mNG and mTFP1.

The following evidence suggest that the observed distribution of lifeact-mTFP1 and dynamin 2-mTFP1 is not significantly affected by FRET. For lifeact-mTFP1 labeling, our main findings are that lifeact-mTFP1-labeled filamentous F-actin grew at the tip of PH-mNG-labeled Λ-profile (e.g., Fig. 1f, g) and that F-actin surrounded Λ-profile's base (Fig. 3f). In both cases, lifeact-mTFP1-labeled F-actin did not overlap with PH-mNG-labeled plasma membrane: most mTFP1-labeled actin filaments were above the tip of the mNG-labeled Λ-profile (Fig. 1f, g), and mTFP1-labeled F-actin ring was more than 60 nm periphery of mNG-labeled Λ-profile's base (Fig. 3f). Thus, our main finding about F-actin distribution should not be influenced by the FRET.

For dynamin 2-mTFP1 labeling, our main findings are that dynamin 2-mTFP1 is located above the tip of the PH-mNG-labeled Λ-profile (Fig. 3b), at the periphery of Λ-profile's base (Fig. 3d), and at the pore region of the Ω-profile (Fig. 4e). Since dynamin 2-mTFP1 puncta are above the tip of the mNG-labeled Λ-profile (Fig. 3b), there should be no FRET from mNG to influence the observation of mTFP1. Since dynamin 2-mTFP1 puncta are in general not overlapped with strong mNG-labeled Λ-profile's base (Fig. 3d), the observation that dynamin-mTFP1 surrounds Λ-profile's base should not be significantly influenced by FRET from strong nearby (<10 nm) mNG.

For dynamin 2-mTFP1 puncta at the mNG-labeled Ω-profile's pore, we performed new experiments by imaging dynamin 2-mNG and PH-mTFP1, in which dynamin 2-mNG fluorescence cannot be absorbed by nearby PH-mTFP1, and thus should not be affected by FRET. We found that dynamin 2-mNG was associated with PH-mTFP1-labeled Ω-profile's pore region (n = 10 events, Supplementary Fig. 8a), similar to the association of dynamin 2-mTFP1 with PH-mNG-labeled Ω-profile's pore region (n = 15, Supplementary Fig. 8b). We also observed that dynamin 2-mNG was associated with PH-mTFP1-labeled Λ-profile (n = 9 events, Supplementary Fig. 8c), similar to the association of dynamin 2-mTFP1 with PH-mNG-labeled Λ-profile (e.g., Fig. 3b, d). We concluded that FRET does not significantly influence our conclusion that F-actin and dynamin are physically available near Λ- and Ω-profile to generate pulling and constriction forces.

**STED scanning modes and recording time**

$XZ/Y_{fix}$ *scanning*. STED images were acquired at the cell bottom at XY (parallel to the coverslip) or XZ (perpendicular to the coverslip) scanning mode (Fig. 1a). Most experiments were performed at the $XZ/Y_{fix}$ scanning mode at the cell bottom, at which images were acquired every 26–200 ms at 15 nm per pixel in an XZ area of 15–20 μm x 0.7–2.5 μm, with a fixed Y-axis location at about the cell center (e.g., Figs. 1b, c and 3b). The imaging duration was limited to 5–20 s before, and 60 s after depol$_{1s}$. We limited to 60 s, because whole-cell endocytosis after depol$_{1s}$,

measured with capacitance recordings, usually takes place within 60 s (e.g., Fig. 1a)[23–25]. Each cell was subjected to only 1 depol$_{1s}$ to avoid endocytosis rundown[55].

$XZ/Y_{stack}$ *scanning*. To reconstruct 3-D structure of endocytic structures, XZ images (X-axis, 12–15 μm; Z-axis, 1–2 μm) were also acquired alone Y-axis every 50 nm for 1–10 μm ($XZ/Y_{stack}$ scanning, collected within ~5–60 s).

*Resolution*. The STED resolution for imaging $PH_G$ in our conditions was ~60 nm on the microscopic X- and Y-axis (parallel to cell-bottom membrane or coverslip), and ~150–200 nm on the microscopic Z-axis. STED images were deconvolved using Huygens software (Scientific volume Imaging) and analyzed with Image J and LAS X (Leica).

**Identifying Flat, Λ, $Ω_p$, $Ω_{np}$ and O with STED XZ imaging**. Identification of Flat, Λ, $Ω_p$, and $Ω_{np}$ with STED XZ imaging was described in detail recently[18]. Λ and Ω were distinguished as below: if a pit's width was the largest at the base, it was Λ-shape; if the pit's width was the largest at the middle of its vertical length, it was a Ω-shape. Λ was identified when the ratio between Λ's height (h) and base (b), h/b, was ≥0.2. Due to the limit of the signal-to-noise ratio, Λ with h/b <0.2 was more difficult to recognized and thus was not taken as Λ, but as flat membrane (Flat). Similarly, due to the detection limit, only those Λ with a b >200 nm were taken as Λ; otherwise, it was defined as Flat. The Ω-shape profile with a visible pore ($Ω_p$) was defined as the $PH_G$-labeled Ω-shape membrane profile with a visible pore (>60 nm, the STED detection limit), which should be less than the Ω-profile width. If a $PH_G$-labeled Ω-profile with a non-visible pore contained A532 spot, it was defined as $Ω_{np}$ (an Ω-profile with a non-visible pore), because the A532 spot indicated an open-pore permeable to A532[25].

**Pore closure identification with STED $XZ/Y_{fix}$ imaging**. During STED $XZ/Y_{fix}$ imaging, A532 was excited at a high laser power so that fluorescent A532 can be bleached with a time constant of 1.5–3.5 s. Pore closure was identified as the gradual dimming of the A532 spot fluorescence to baseline during $XZ/Y_{fix}$ PH-EGFP/A532 imaging. A532 fluorescence dimming is due to pore closure that prevents bleached A532 (by strong excitation) from exchange with a large reservoir of fluorescent A532 (very small molecule, ~1 nm) in the bath. Pore closure detected by this bleaching method was confirmed by four sets of evidence. First, upon fusion pore closure detected with the bleaching method, bath application of an acid solution could not quench the pH-sensitive VAMP2-EGFP or VAMP2-pHluorin expressed at the fused vesicle, suggesting that fusion pore closure prevents the exchange of $H^+$ and $OH^-$, the smallest molecules between the 'closed' vesicle and the bath solution. Second, fusion pore closure detected with the bleaching method can be blocked by dynamin inhibitors or dynamin knockdown, which blocks endocytosis[23,25]. Third, adding all events of fusion pore closure and closure of the preformed Ω's pore detected with the bleaching method closely predicts ultrafast, fast, slow or no endocytosis detected with capacitance measurements at the same cell[18]. Preformed Ω's pore closure detected with the bleaching method could be blocked by dynamin inhibition, suggesting that it is mediated by dynamin[18].

**Calculation of transition probabilities**. The probability for Λ to make Λ→Ω [Prob(Λ)$_{→Ω}$] was calculated as the number of Λ→Ω divided by the total number of Λ per scanning region during $XZ/Y_{fix}$ scanning. Similarly, the probability for Ω to make Ω→O [Prob(Ω)$_{→O}$] was calculated as the number of Ω→O divided by the total number of Ω per scanning region during $XZ/Y_{fix}$ scanning. Flat's probability to make Flat→Λ was calculated as the number of Flat→Λ transitions per unit of Flat ($N_{Flat→Λ}$), which was arbitrarily defined as 10 μm. Thus, $N_{Flat→Λ}$ reflected the frequency of Flat→Λ transition along X-axis during $XZ/Y_{fix}$ scanning (X-axis length per $XZ/Y_{fix}$ scanning: 16.1 ± 0.1 μm, n = 513 cells).

$N_{Flat→Λ}$, Prob(Λ)$_{→Ω}$ and Prob(Ω)$_{→O}$ were negligible when measured in the absence of depol$_{1s}$ (Fig. 4 in ref. [18]). $N_{Flat→Λ}$, Prob(Λ)$_{→Ω}$ and Prob(Ω)$_{→O}$ measured after depol$_{1s}$ were much higher in cells with ICa >300 pA than with ICa <300 pA (Fig. 4g in ref. [18]). Furthermore, $N_{Flat→Λ}$, Prob(Λ)$_{→Ω}$ and Prob(Ω)$_{→O}$ measured after depol$_{1s}$ were substantially reduced when extracellular calcium was replaced with strontium (Fig. 4h in ref. [18]). These results indicate that Flat→Λ, Λ→Ω and Ω→O are triggered by calcium influx, excluding the possibility that they are STED-laser-induced artifacts.

**Electron microscopy on liposomes incubated with dynamin**. 1,2-dioleoyl-sn-glycero-3-phospho-L-serine (100 μl of 5 mg/ml, DOPS, Avanti) was dried and resuspended in 250 μl HCB150 (50 mM HEPES, 150 mM KCl, 2 mM EGTA, 1 mM MgCl₂, 1 mM TCEP, pH 7.5). Unilamellar liposomes were obtained by extruding the mixture 21 times through a 1 μm pore-size polycarbonate membrane (Avanti). Recombinant full-length dynamin 1 was purified from mammalian 293-F cells[30,56] and ΔPRD-dynamin 1 was purified from Sf9 insect cells (ThermoFisher Scientific). Briefly, recombinant baculovirus containing the sequence of ΔPRD-dynamin 1 with an N-terminal His-tag was generated by following Bac-to-Bac Baculovirus Expression System (ThermoFisher Scientific). The suspension cultures of Sf9 were maintained in Sf-900 III serum-free media (SFM, ThermoFisher Scientific) and inoculated with recombinant baculovirus at a cell density of $1.6 \times 10^6$ with 1/10

volume of virus/final volume of the medium. The cells were grown for 72 hours at 27 °C, and pelleted by centrifugation at 1000 × g, 5 min, 4 °C. The pellet was resuspended in modified HSB150 (50 mM HEPES, 150 mM KCl, 5 mM β- mer-captoethanol,10 mM Imidazole, pH 8.0) and containing protease inhibitor cocktail (Millipore Sigma). The cells were then lysed by sonication (total time of 8 min with 5 sec pulse-on and 15 sec pulse-off) followed by high speed centrifugation (20,000 × g, 15 min). The supernatant was collected, passed through Ni-NTA beads and the protein was eluted with 150 mM imidazole in modified HSB150. The protein solution was dialyzed in HSB150 overnight and the purity was checked using SDS-PAGE/Coomassie staining.

Dynamin decorated tubes were generated by incubating liposomes with protein (0.5 mg/ml, in HCB150) for 1 min, 5 min or 1 hour. 3.5 μl of each sample was applied to the plasma-cleaned C-flat grids (Electron Microscopy Sciences, CF-1.2/1.3–4Au-50), pre-blotted, and blotted with filter paper (Whatman 1) for 2 sec followed by plunging into liquid ethane using a Leica EM GP (Leica Microsystems). The samples were stored in liquid nitrogen until data acquisition. Data was collected on a TF20 microscope (ThermoFisher Scientific) at 200 kV and the imaging was done at a nominal magnification of ×29,000 with a defocus range of 1.5–3.0 μm on a K2 summit camera (Gatan) in counting mode. Motion correction and dose weighting were done by MotionCor 1.2.1[57]. The diameter measurement of samples was calculated in Fiji[58].

For cryo-electron tomography, tilt series were collected on a 200 kV Thermo Scientific Glacios transmission electron microscope at 17,500× nominal magnification (calculated pixel size is 2.37 Å), with a target defocus of -15.0 μm using a K3 direct detection camera (Gatan. Inc.). Tilt series were recorded using SerialEM[59] software at angles between −60° to +60° with a tilt increment of 3° (12 frames/image). Tomograms were reconstructed from the tilt series with Etomo in IMOD[60]. Segmented tomograms were generated using the EMAN2 semi-automated segmentation package[61] and were analyzed in ChimeraX[62].

**Data selection and statistics.** The data within the first 2 min after whole-cell break-in were used to avoid whole-cell endocytosis rundown[25,55]. Cells expressed with PH$_G$, PH-mTFP1, or PH-EGFP were used for imaging of membrane dynamics. The statistical test used is $t$ test. The data were expressed as mean ± s.e.m. Each group of data was from at least four primary chromaffin cell cultures. Each culture was from 3–4 glands from 2 bovines.

**Hippocampal culture.** Mouse hippocampal culture was prepared from Wild-type (C57BL/6J) or Actb$^{LoxP/LoxP}$ mice at P0 of either sex[63,64]. Hippocampal CA1-CA3 regions were dissected, dissociated, and plated on Poly-D-lysine treated coverslips. Cells were maintained at 37 °C in a 5% CO$_2$ humidified incubator with a culture medium consisting of Neurobasal A (Invitrogen, Carlsbad, CA), 10% fetal bovine serum (Invitrogen, Carlsbad, CA), 2% B-27 (Invitrogen, Carlsbad, CA), 0.25% glutamax-1 (Invitrogen, Carlsbad, CA), and 0.25% insulin (Sigma, St. Louis, MO). On 5–7 days after plating, neurons were transfected with plasmids using Lipo-fectamine LTX (Invitrogen, Carlsbad, CA).

**Electron microscopy at hippocampal cultures.** Electron microscopy was performed to examine ultrastructural changes in Cre-ER$^{TM}$; Actb$^{LoxP/LoxP}$ hippocampal cultures at 4 days after 4-OH-tamoxifen (1 μM) treatment, which reduced β-actin to ~13% of control[16]. Hippocampal cultures were fixed with 4% glutar-aldehyde (freshly prepared, Electron microscopy sciences, Hatfield, PA) in 0.1 N Na-cacodylate buffer solution containing for at least one hour at 22–24 °C, and stored in 4 °C refrigerator overnight. The next day, cultures were washed with 0.1 N cacodylate buffer, and treated with 1% OsO$_4$ in cacodylate buffer for 1 hr on ice, and 0.25% uranyl acetate in acetate buffer at pH 5.0 overnight at 4 °C, dehydrated with ethanol, and embedded in epoxy resin. Thin sections were counterstained with uranyl acetate and lead citrate then examined in a JEOL 200 CX TEM. Images were collected with a CCD digital camera system (XR-100 from AMT, Danvers, MA) at a primary magnification of ×10,000–20,000. Synapses were selected based on the structural specialization including synaptic vesicle clustering, synaptic cleft and the postsynaptic density.

**Measurements and data analysis at hippocampal cultures.** In electron microscopy, pits, including Λ and Ω, were defined as having a height >15 nm, a base of 20–120 nm, and a height/base ratio >0.15. Λ and Ω were distinguished base on their shapes. If the pit's width was the largest at the base, it was Λ-shape; if the pit's width was the largest at the middle of its vertical length, it was a Ω-shape. Means were presented as ± s.e.m. The statistical test was two-tailed unpaired $t$-test. Each culture was from 3–6 mice. Each group of data was obtained from at least four batches of cultures (4–7 cultures).

**Theoretical modeling**

*Brief summary of physical modeling and image processing.* In order to model the shape of the endo-site membrane under specific spatial constraints we numerically minimized the Helfrich's elastic energy[65] of the membrane,

$$F = \int_M \left(\frac{\kappa}{2}J^2 + \gamma\right)dA - \pi r_b^2 \gamma \qquad (1)$$

where the integration was done over the surface of the invagination, $J$ is the mean curvature, $\kappa$ is the membrane bending modulus $\kappa = 0.8 \cdot 10^{-19}$ Joule[66]; $\gamma$ is the lateral tension in the plasma membrane surrounding the endo-site; $r_b$ is the radius of the endo-site base. We solved the problem under the assumption of rotational symmetry, hence, representing the membrane shape in cylindrical coordinates. For computations we used Brakke's "Surface Evolver"[67].

The directly computed value was the membrane energy, $F$, as a function of the endo-site height, $H$, and base radius, $r_b$. The pulling force, $f_{pull}$, and the edge-generated constriction force, $f_{constrict}$, were computed as derivatives of the energy with respect to $H$ and $2\pi r_b$, respectively.

The analysis of the system was done in the dimensionless parameter space with all length values normalized by the intrinsic length $r_i = \sqrt{\frac{\kappa}{2\gamma}}$, and the pulling force normalized by the intrinsic force, $f_i = \sqrt{2\kappa\gamma}$, after which the non-normalized energy and force values were recovered using the above value of the bending modulus.

Processing of the videos of the endocytosis events was performed by fitting the membrane profile observed at each frame to a computed membrane shape by using three fitting parameters, the base radius, $r_b$, the height, $H$, and the tension, $\gamma$. The fitting procedure was done using MATLAB tools (MATLAB R2019a, MathWorks) by first skeletonizing the images using medial axis transform, and then finding the fitting parameter values, which provided the minimal sum of squared distances of the skeleton to the computed shape profile. Each frame was analyzed for determination of physical properties of the system such as forces applied to the membrane. Finally, the experimentally observed and the corresponding computed shape profiles were stacked for a dynamic representation of the shape evolution.

*Computational model.* We consider the endo-site as a circular fragment of the plasma membrane bounded by a ring-like proteinic structure. In the initial state, the plasma membrane and, hence the endo-site membrane are flat and subjected to a lateral tension, $\gamma_0$. The endo-site membrane is assumed to have no protein coat on it and to possess mechanical properties typical to those of a lipid bilayer, i.e., to exhibit a lateral fluidity and a resistance to bending deformations quantified by a bending modulus, $\kappa$.

Membrane budding is proposed to be driven by application to the endo-site center of a local force, $f_{pull}$, pulling the membrane towards the cell interior (Supplementary Fig. 6a). In the course of budding, unless extra discussed, the endo-site membrane is assumed to be able to exchange the area with the flat plasma membrane through the smooth membrane connection at the endo-site boundary characterized by the radius, $r_b$, and referred below to as the endo-site base.

We consider the equilibrium shapes adopted by the endo-site membrane to correspond to the minimal value of the elastic energy, $F_{el}$, upon given values of the parameters $f_{pull}$, $\gamma_0$, $\kappa$, and $r_b$. The elastic energy has two contributions, the energy of membrane bending, $F_B$, and that of membrane tension, $F_\gamma$. The bending energy is provided by Helfrich model[65]. We assume that the membrane is symmetric and characterized, therefore, by a vanishing spontaneous curvature, and that the membrane does not change its connectivity in the course of budding, which means that we do not include in our analysis the membrane fission event completing the formation of an endocytic vesicle. In this case, the local surface density of the bending energy is given by[65]:

$$f_B = \frac{\kappa}{2}J^2 \qquad (S1)$$

where $J$ is the surface mean curvature[68] such that the total bending energy is given by integration of $f_B$ over the area of the endo-site membrane,

$$F_B = \oint f_B dA \qquad (S2)$$

The energy of the tension computed with respect to the initial flat state is given by

$$F_\gamma = \gamma_0\left(A - \pi r_b^2\right) \qquad (S3)$$

where $A$ and $\pi r_b^2$ are the areas of the endo-site membrane in the budded and the initial flat state, respectively.

Minimization of the total elastic energy is performed under the following assumptions. First, we consider the endo-site shapes to be rotationally symmetric with respect to the central axis, $z$, such that a shape is described by its cross-sectional profile characterized at each point by the local tangent angle, $\phi$, which depends on the radial coordinate, $r$ (Supplementary Fig. 6a). The smooth connection of the endo-site membrane to the flat plasma membrane is accounted by the boundary condition, $\phi(r = r_b) = 0$. Second, to avoid a non-physical singularity of the bud profile in its center, we assume the pulling force, $f_{pull}$, to be applied not point-wise but rather along a small disk-like membrane insertion, the "plug", whose center is located at the z-axis ($r = 0$) and the radius, $r_{plug}$, is much smaller than that of the base, $r_{plug} \ll r_b$. The value of the tangent angle at the plug boundary, $\phi(r = r_{plug})$, depends on the unknown details on the membrane-plug interaction. We performed pilot computation for two limiting cases, namely, assuming a fully flexible membrane-plug connection such that $\phi(r = r_{plug})$ can

adopt any value, and considering a smooth transition between the membrane and the plug, $\phi\left(r = r_{plug}\right) = 0$. The computation results appeared to be rather insensitive to this assumption so that in the main calculations we adopted the latter boundary condition, $\phi\left(r = r_{plug}\right) = 0$.

To perform the computations, we use the relationship between the pulling force, $f_{pull}$, and the distance, $H$, between the initial membrane plane and the point of the force application (Supplementary Fig. 6a), which will be referred below to as the height of the bud. Fundamentally, a force can be computed as a derivative of the energy with respect to the associated displacement. Based on that, we use as geometrical parameters, setting the membrane configuration, the base radius, $r_b$, and the bud height, $H$. We perform the energy minimization and find the equilibrium shapes and the corresponding minimal energies, $F_{el}^*$, of the endo-site membrane for different sets of the geometrical parameters, $r_b$, and $H$. The resulting energy function, $F_{el}^*(r_b, H)$, is then used for calculation of the pulling force corresponding to each value of the based radius, $r_b$, according to, $f_{pull}\left(r_b, H\right) = \frac{\partial F_{el}^*(r_b, H)}{\partial H}$. The resulting relationship is then used to present the shapes and the energies as functions of the experimentally relevant parameters, $f_{pull}$ and $r_b$.

Before turning to the numerical computations, we take advantage of the scaling considerations according to which the system is characterized by a characteristic internal length scale

$$r_i = \sqrt{\frac{\kappa}{2\gamma_0}} \qquad (S4)$$

and force scale,

$$f_i = \sqrt{2\kappa\gamma_0}$$

The membrane shapes predicted by the model and the corresponding elastic energy depend on the normalized values of the geometrical parameters, $\frac{H}{r_i}$ and $\frac{r_b}{ri}$, and of the pulling force, $\frac{f_{pull}}{f_i}$. Therefore, we will present the results of the analysis in terms of the normalized parameters.

The computations, which included variations of the endo-site membrane shapes, calculations of the corresponding energies, and determination of the minimal energy membrane configuration for every parameter set, were performed numerically using Ken Brakke's "Surface Evolver" based on the Finite Element approach.

### Classification of endo-site membrane configurations and the shape diagram.
Computation of the equilibrium shapes with different parameter sets retrieves various types of membrane configurations. Keeping in mind our experimental observations, we classified the computed membrane shapes into three qualitatively different categories: (i) $\Lambda$-shapes, (ii) $\Omega$- shapes, and (iii) tether-shapes. As a criterion for this classification we used the maximal value, $\phi^*$, reached by the tangent angle, $\phi$, along the shape profile (Supplementary Fig. 6a).

We define a membrane configuration as $\Lambda$-shape if it is so flat that $\phi^* < 82°$; configurations whose low part has an hour-glass-like neck such that $\phi^* > 98°$ are defined as $\Omega$-shapes; finally, the intermediate configurations with a nearly cylindrical upper part parallel to z-axis such that, $82° < \phi^* < 98°$, belong to the tether-shapes. The $16°$ range of the angle $\phi^*$ defined to encompass the tether shapes is somewhat arbitrary and based on the visual ability to recognize deviations of line orientation from the vertical axis.

Examples of the computationally obtained $\Lambda$- and two $\Omega$- shapes are presented in the main text (Fig. 7b). The energies of the computed membrane configurations are presented as energy landscape, $F_{el}(\frac{H}{r_i}; \frac{r_b}{r_i})$, in (Supplementary Fig. 6b). The results of the entire computations are accounted for by the shape diagram (Fig. 7b), which presents the ranges of the geometrical parameters, $\frac{H}{r_i}$ and $\frac{r_b}{ri}$, for which the computed shapes belong to each of the three categories defined above. A more general diagram, which does not use the criterion of the configuration type but rather presents the maximal tangent angle, $\phi^*$, for different combinations of the geometrical parameters, $\frac{H}{r_i}$ and $\frac{r_b}{r_i}$, is presented in Supplementary Fig. 6c.

### Membrane shape evolution.
Our experiments showed a series of membrane shapes corresponding to time evolutions of the endo-site membrane configurations. These images enabled us to propose that the distinct stages of the shape evolutions are driven by changes of different system parameters. Specifically, the stage of transformation of the flat shape of the endo-site membrane into the $\Lambda$-configuration with continuously growing height, $H$, the Flat→$\Lambda$ transition, is driven by an increasing pulling force, $f_{pull}$, upon a nearly constant radius of the endo-site base, $r_b$. The next stage, the $\Lambda$→$\Omega$ transition, and the narrowing of the $\Omega$-shape neck, result, primarily, from constriction of the base, i.e. decrease of the base radius $r_b$.

Based on that we modeled the membrane shape sequence corresponding to Flat→$\Lambda$ transition by assuming a fixed value of the base radius, $\frac{r_b}{r_i} = 2.5$, and a gradually increasing pulling force, $f_{pull}$. The endo-site membrane was assumed to freely exchange the material with the plasma membrane, which played a role of a lipid reservoir with tension $\gamma_0$, which resulted in a continuous increase in the endo-

site area. The results are presented in the main text (Fig. 7c). A three-dimensional representation of the membrane configurations emerging in the course of Flat→$\Lambda$→$\Omega$ transition is shown in Supplementary Fig. 7a.

To model the $\Lambda$→$\Omega$ transition we assumed that the normalized base radius, $r_b$, decreases, while the pulling force, $f_{pull}$, remains constant and equal to its value at the end of the Flat→$\Lambda$ shape sequence. Our analysis demonstrated that, to explain the shape sequence experimentally observed for this transition, we need to make an additional assumption concerning the exchange of the membrane material between the endo-site and the plasma membrane. The reason for that is the structure of the computed energy landscape, which demonstrates that in case the membrane is freely exchangeable between the endo-site and plasma membranes during the $\Lambda$→$\Omega$ transition, decrease of $r_b$ upon a constant $f_{pull}$ would lead to decrease of the bud height, $H$, and ultimate absorption of the bud by the plasma membrane instead of development of $\Omega$-shape with a narrowing neck. Therefore, we assumed that the area of the endo-site membrane accumulated at the end of the Flat→$\Lambda$ transition is kept constant during the $\Lambda$→$\Omega$ transformation, which means that no exchange occurs between the endo-site and the plasma membrane. A plausible reason for this condition is an effective dynamic friction between the membrane at the endo-site boundary and the underlying proteinic structure, which must restrict the membrane flow through the boundary. In case the base constriction is sufficiently fast, the limited membrane flow would not enable the membrane area equilibration between the endo-site and plasma membranes, which, in turn, would mean an effective conservation of the endo-site area within the time scale of the base constriction. The results for the computed shape sequence are presented in the main text (Fig. 7d).

It is important to add that, once the fast $\Lambda$→$\Omega$ transition occurred upon a constant membrane area, the resulting $\Omega$-shape of the endo-site membrane shape is predicted to undergo no or only very slow relaxation. The reason is that, according to the computed energy landscape, the transition of the reached $\Omega$–shape into the state corresponding to the endo-site absorption by the plasma membrane would require overcoming by the system of a large energy barrier and, hence, practically unfeasible.

### Fitting of the computed membrane shapes to the observed profiles.
To verify our model, we asked whether the computed shapes of the endo-site membrane obtained for different stages of the budding process reproduce the experimentally observed membrane profiles. A good quantitative agreement between the details of the experimental and computational membrane shapes would be a strong argument in favor of the correctness of our mechanism.

To explore this issue, we performed fitting of the computed membrane profiles to the sequences of profiles constituting the recorded videos of the Flat→$\Lambda$→$\Omega$ transitions. First, we processed the videos to present the observed smeared profiles as series of points. This was done frame by frame using the MATLAB image processing and optimizing tools. Each frame of a video was processed individually by skeletonizing, using the medial axial transform, the fluorescent image describing the membrane. Second, for each frame we fitted the obtained experimental point series by a shape profile given by our model. The fitting was performed by minimizing the sum of squared distances between the experimental points and the model profile by using, as the fitting parameters, the profile height, base radius, $r_b$, and the intrinsic length, $r_i$ (Eq. S4).

The processed experimental points and the corresponding fitted computational curve are shown in (Supplementary Fig. 7b, c) for selected frames of two experimental videos (Fig. 7e, f), one corresponding to Flat→$\Lambda$ (Supplementary Fig. 7b) and the second to $\Lambda$→$\Omega$ (Supplementary Fig. 7c) transition. The complete series of fitted profiles are presented as computational movies, whose direct comparison with the experimental videos provides an additional substantiation of our model, as presented in the main text.

Based on the fitted values of the parameters, $r_b$, H and $r_i$ (Eq. S4) determined for each video frame, we calculated the corresponding values of the forces acting in the system: the pulling force, $f_{pull}$, and the membrane tension, $\gamma_0$. These calculations were done using a conventional value for the bending rigidity of the membrane[66], $\kappa = 0.8 \cdot 10^{-19} J$. The values of $f_{pull}$ presented in the main text (Fig. 7e) were in the range 1–3 pN, which corresponds to the forces exerted by just few molecular motors such as myosin-actin complexes[69,70]. The fitted tension values were around $\gamma_0 \cong 1\frac{\mu N}{m}$, which corresponds to the lowest tension level measured in live cells[71].

**Reporting summary.** Further information on research design is available in the Nature Research Reporting Summary linked to this article.

## Data availability
The datasets generated during and/or analyzed during the current study are available from the corresponding authors on reasonable request. Source data are provided with this paper.

## Code availability
The main computational codes used for simulations and fitting procedures in this paper are available at https://github.com/benzucker-tau/Endocytosis.

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

## Acknowledgements

We thank Tamas Balla for providing PH-EGFP, Carolyn Smith for STED microscopy support, Susan Cheng (NINDS) for EM support, John Jimah (NIDDK) for full-length-dynamin 1 protein, Haifeng He and Yanxiang Cui (NIDDK) for cryoEM data collection, and the NIDDK EM Core Facility. This work was supported by NINDS and NIDDK Intramural Research Programs (ZIA NS003009-13 and ZIA NS003105-08; Z01 DK060100). M.M.K. was supported by SFB 958 "Scaffolding of Membranes" (Germany), Singapore-Israel (NRF-ISF) research grant 3292/19.

## Author contributions

W.S. performed and analyzed most live-cell experiments, and participated in designing experiments; B.Z. performed theoretical modeling; N.K. performed cryoEM on dynamin in liposomes; B.S. and S.H.L. performed EM works in hippocampal cultures; X.G., C.Y.C., and J.M.T. helped performing live-cell works. J.T.H. helped performing cryoEM work. J.E.H. supervised and described cryoEM work with help from N.K.; M.M.K. supervised and described modeling work with help from B.Z.; L.G.W. supervised live-cell works, EM works on hippocampal cultures, designed the project and wrote the manuscript with help from all authors.

## Funding

## Competing interests

The authors declare no competing interests.
