## [Peer Review File · Nature Communications]

REVIEWER COMMENTS

Reviewer #1 (Remarks to the Author):

In this manuscript, Wonchul Shin et al examined how non-coated-membrane budding is mediated by actin filaments and dynamin through STED microscopy imaging and physical modeling. They found polymerized actin filaments and dynamin generate a pulling force transforming flat membrane into Λ -shape, dynamin helices surround and constrict Λ -profile's base, converting Λ - into Ω -shape, and then constrict Ω -profile's pore until a vesicle is formed. Overall, the manuscript is interesting and convincing, and may have some potential influences on many membrane shaping processes. However, the following concerns must be addressed.

1. The authors should explain how actin generates pulling force and whether myosin is involved. Furthermore, whether other cytoskeleton proteins, especially microtubule that can generates pulling force in cells, incorporate the non-coated-membrane budding process, should be discussed to some extend.
2. In the model section, the authors didn't cite any reference. However, many researchers did many classical investigations in the area of membrane budding and tubule formation. The authors should not neglect the contributions of these previous works.

Reviewer #2 (Remarks to the Author):

In the manuscript titled "Molecular mechanics underlying flat-to-round membrane budding in live secretory cells", Wonchul Shin and colleagues study the mechanism of clathrin-independent and dynamin-dependent endocytosis in chromaffin cells. They propose that membrane curvature and budding formation is mediated by actin and dynamin, while posterior membrane constriction and vesicle closure is driven by dynamin. This is a very detailed and thorough research article, well written and easy to understand, technically rigorous and with very interesting findings. The strength of Dr Wu's research approach is that it monitors the endocytic process in real time, allowing the direct observation of the steps and how they are modified by pharmacological or genetic manipulations. I do not think additional experiments are necessary, but I have concerns about the interpretation of the results and the discussion of the existing bibliography, as detailed below.

1. Previous work from Dr Wu's laboratory has shown that actin and dynamin play opposing roles during exocytosis in chromaffin cells. Actin facilitates pore expansion while dynamin mediates pore constriction. How do the authors reconcile these roles during pore formation and expansion with the current findings on membrane budding and pore closure? Particularly regarding actin, how does actin drive pore expansion and at the same time generates an inward pulling force for endocytosis? This ought to be discussed.

2. At the end of the introduction the authors state that "current models [...] may need to be re-examined and modified to account for the powerful transformation forces of dynamin and actin reported here". I wish the authors would have cited and discussed the work of many laboratories that have been studying clathrin-independent endocytosis in the last couple of decades. Previous research in different cell types have shown that clathrin-independent endocytosis is mediated by pulling forces generated by actin working together with BAR-domain containing proteins and/or dynamin (dynamin is not always essential, since there are a number of dynamin-independent endocytic mechanisms). To keep my comment brief, I will only refer to two well written review articles: Mayor, Satyajit et al. "Clathrin-independent pathways of endocytosis." Cold Spring Harbor perspectives in biology vol. 6,6 (2014) a016758; and Ferreira, Antonio P A, and Emmanuel Boucrot "Mechanisms of Carrier Formation during Clathrin-Independent Endocytosis." Trends in cell biology vol. 28,3 (2018): 188-200. Thus, in this context, the current finding that actin and dynamin mediate flat to invaginated transitions is not so "unexpected" as the authors write at the beginning of the discussion, but it corroborates previous discoveries in other systems. The strength of the present work is that the authors follow the budding process in real time using super-resolution microscopy, a technique in which this lab is an expert. I think the authors should give credit to earlier work from others and discuss previous discoveries in the endocytic field beyond neuroendocrine systems.

3. How do the authors reconcile the current finding of dynamin's role in forming flat to invaginated transitions, with the previous observations of multiple invaginations and long tubules in dynamin knock-out cells? These well-known older electron microscopy observations were interpreted as endocytic structures that failed to be excised, thus suggesting that dynamin is not essential for pit formation.

4. The authors do a great job at quantifying the occurrence of all the transitions studied, this is very valuable since quantitative microscopy is sometimes overlooked in the field. The transitions from Λ to Ω and from Ω to O have a very low probability (0.12 and 0.24, respectively). What is the interpretation of this finding? What happens in the remaining ~ 0.8 cases, the pit collapses back to flat and there is no endocytosis? What is the biological relevance of having such a stochastic mechanism for endocytosis?

Reviewer #3 (Remarks to the Author):

In this manuscript, Shin, Zucker and Kundu et al showed the role of dynamin and filamentous actin on membrane budding process using STED imaging. Overall, it is a very well-written manuscript with well-performed experiments. In my opinion, this paper should be published in Nature Communications.

I mostly evaluated the imaging and membrane aspect of the manuscript. My minor comments are below.

Do the authors have control experiments where the budding cannot occur (e.g. no Calcium). This should rule out STED-induced tubulation artefacts.

Do authors have control experiments ruling out FRET between different fluorophores that can bias the results on protein distributions.

Laser powers used for STED imaging should be given as absolute numbers (microwatt, milliwatt), not as % of the total powers where the total powers are not mentioned.

xz images might be interpreted as either 3D image with 2D-STED or 3D-STED. It should clearly be mentioned whether the authors have done 2D STED using 2D doughnut, or z(3D)-STED using 3D depletion pattern.

Red/green image combination is impossible to read for color-blind readers, I would consider changing red to magenta.

Point-to-point response
(Reviewers' comments are in italic)

Reviewer: 1

In this manuscript, Wonchul Shin et al examined how non-coated-membrane budding is mediated by actin filaments and dynamin through STED microscopy imaging and physical modeling. They found polymerized actin filaments and dynamin generate a pulling force transforming flat membrane into Λ -shape, dynamin helices surround and constrict Λ -profile's base, converting Λ - into Ω -shape, and then constrict Ω -profile's pore until a vesicle is formed. Overall, the manuscript is interesting and convincing, and may have some potential influences on many membrane shaping processes. However, the following concerns must be addressed.

Reply: We appreciate the reviewer's strong support of our work.

Major comments:

1. The authors should explain how actin generates pulling force and whether myosin is involved. Furthermore, whether other cytoskeleton proteins, especially microtubule that can generate pulling force in cells, incorporate the non-coated-membrane budding process, should be discussed to some extent.

Reply: As the reviewer suggested, we discuss how actin generates pulling forces and speculate on the potential participation of myosin in the Discussion section as in the following.

Page 20, 2nd paragraph: "Our findings raise the questions as to how F-actin filaments generate a pulling force. One possibility is that it is generated by network of actin filaments that surround the membrane bud and undergoes polymerization at the bud base^{32-34, 36}. The force resulting from polymerization could be transmitted to the bud apex³³. Alternatively, the pulling force can be generated by a bundle of contractile complexes of actin and myosin filaments⁴¹, the stress-fibre⁴², whose one end is attached to the bud tip, whereas the second end is anchored in the cytoplasmic actin network. Dynamin could mediate the actin filament bundling within the stress fibre^{30, 43}. Given a relatively large size of the endocytic buds studied here, and the observation that actin is attached to Λ 's tip and the associated spike-like protrusions, we favour the latter mode, i.e., the point force generation."

2. In the model section, the authors didn't cite any reference. However, many researchers did many classical investigations in the area of membrane budding and tubule formation. The authors should not neglect the contributions of these previous works.

Reply: We thank the reviewer for commenting on this flaw of us, which happened at some stage of the text condensing that initially contained all the references. As the reviewer suggested, we cited the major works on the budding modelling in the Methods section (for detail see pages 37-42 where we cited Refs. 65-71). We also included a few sentences describing and citing additional important studies (Refs. 31-36) about membrane budding in the Results section in the revised manuscript as in the following.

Page 15, 1st paragraph: "Our computational model synthesized elements of previous models aimed at theoretical analysis of endocytic bud formation and scission upon various conditions in yeast and mammalian cells³¹⁻³⁶. The specificity of our model is in an explicit accounting for the vesicle formation with constriction of the bud base by a protein structure, and a point force at the bud tip in the absence of protein coat on the bud surface."

Reviewer: 2

In the manuscript titled “Molecular mechanics underlying flat-to-round membrane budding in live secretory cells”, Wonchul Shin and colleagues study the mechanism of clathrin-independent and dynamin-dependent endocytosis in chromaffin cells. They propose that membrane curvature and budding formation is mediated by actin and dynamin, while posterior membrane constriction and vesicle closure is driven by dynamin. This is a very detailed and thorough research article, well written and easy to understand, technically rigorous and with very interesting findings. The strength of Dr Wu’s research approach is that it monitors the endocytic process in real time, allowing the direct observation of the steps and how they are modified by pharmacological or genetic manipulations. I do not think additional experiments are necessary, but I have concerns about the interpretation of the results and the discussion of the existing bibliography, as detailed below.

Reply: We appreciate the reviewer’s strong support of our work.

1. Previous work from Dr Wu’s laboratory has shown that actin and dynamin play opposing roles during exocytosis in chromaffin cells. Actin facilitates pore expansion while dynamin mediates pore constriction. How do the authors reconcile these roles during pore formation and expansion with the current findings on membrane budding and pore closure? Particularly regarding actin, how does actin drive pore expansion and at the same time generates an inward pulling force for endocytosis? This ought to be discussed.

Reply: As the reviewer suggested, we added a paragraph discussing these apparently different roles of F-actin as in the following.

Page 20, last paragraph – page 21, 1st paragraph: “The synergy between F-actin and dynamin in endocytic membrane invagination (Flat→ Λ) is in contrast to a recent finding that F-actin facilitates fusion pore expansion whereas dynamin constricts fusion pore and thus counteracts pore expansion in chromaffin cells²⁴. This apparent difference might originate from a qualitative difference between the machineries mediating the actin action in these two processes. Fusion pore expansion is facilitated by an enhanced membrane tension, which can be produced and/or modulated by a force pushing the plasma membrane and resulting from polymerization of cortical actin filaments against the membrane^{24, 44}. In contrast, endocytic membrane invagination may be driven by an actin/dynamin-dependent pulling force applied at Λ ’s tip, as suggested in the present work.”

2. At the end of the introduction the authors state that “current models [...] may need to be re-examined and modified to account for the powerful transformation forces of dynamin and actin reported here”. I wish the authors would have cited and discussed the work of many laboratories that have been studying clathrin-independent endocytosis in the last couple of decades. Previous research in different cell types have shown that clathrin-independent endocytosis is mediated by pulling forces generated by actin working together with BAR-domain containing proteins and/or dynamin (dynamin is not always essential, since there are a number of dynamin-independent endocytic mechanisms). To keep my comment brief, I will only refer to two well written review articles: Mayor, Satyajit et al. “Clathrin-independent pathways of endocytosis.” Cold Spring Harbor perspectives in biology vol. 6,6 (2014) a016758; and Ferreira, Antonio P A, and Emmanuel Boucrot “Mechanisms of Carrier Formation during Clathrin-Independent Endocytosis.” Trends in cell biology vol. 28,3 (2018): 188-200. Thus, in this context, the current

finding that actin and dynamin mediate flat to invaginated transitions is not so “unexpected” as the authors write at the beginning of the discussion, but it corroborates previous discoveries in other systems. The strength of the present work is that the authors follow the budding process in real time using super-resolution microscopy, a technique in which this lab is an expert. I think the authors should give credit to earlier work from others and discuss previous discoveries in the endocytic field beyond neuroendocrine systems.

Reply: As the reviewer pointed out, pharmacological inhibition of actin polymerization has led to inhibition of numerous forms of clathrin-independent endocytosis, suggesting the involvement of actin in clathrin-independent endocytosis, most likely in pit formation. Actin has thus been suggested to be involved in pit formation, as pointed out in many reviews. However, in the absence of core coat-proteins capable of forming vesicle-like cages, what type of physical forces mediate non-coated-membrane budding or clathrin-independent budding remain largely unclear. How F-actin contributes to the generation of these physical forces is thus unclear. By real-time imaging of the flat-to-round membrane transformation, genetic and pharmacological manipulation aiming to identify proteins involved in these transitions, *in vitro* reconstitution of proteins’ unconventional functions and realistic physical modelling, we discovered two types of budding forces underlying non-coated-membrane budding. First, a pulling force mediated by polymerized actin filaments together with the GTPase dynamin (well known as an enzyme mediating fission of ~5 nm narrow necks) transforms flat membrane into Λ -shape (Λ). Second, dynamin helices surround and constrict the Λ -profile’s base, converting Λ into Ω -shape (Ω), and then constrict the Ω ’s pore until a vesicle is formed.

As the reviewer suggested, we rewrote the introduction crediting and discussing earlier works about actin beyond neuroendocrine systems, where we cited two reviews the reviewer mentioned above, an additional review published last year, and 5-6 original studies. We avoid using “unexpected” to describe our finding about actin. We rewrote the last sentence of Introduction to describe more detail of our findings. We also modified the Discussion section discussing that our findings support the early works about actin and provide a more detail mechanical role for actin. Most of these modifications, which should address the reviewer’s concern, are quoted below for the convenience of the reviewer to read.

Page 3, last paragraph – page 4, 1st paragraph: “What type of physical forces mediate formation of non-coated vesicles? Many studies suggest that cytoskeletal filamentous actin (F-actin) is involved⁵⁻⁷. For example, inhibition of actin polymerization impairs several forms of clathrin-independent endocytosis, such as ultrafast endocytosis^{10, 14} and fast endophilin-mediated endocytosis¹⁵ (for review, see Refs. ⁵⁻⁷). Deletion of actin β - or γ -isoform or application of latrunculin A that inhibits F-actin polymerization reduces the pit numbers during synaptic vesicle endocytosis that is considered clathrin-independent^{11, 14, 16, 17}. These results suggest actin involvement in non-clathrin-coated pit formation⁵⁻⁷.”

Last sentence of Introduction, as quoted by the reviewer at the beginning of comment 2, is changed to: “Current models for dynamin/actin-dependent non-coated-membrane budding and coated-membrane budding may need to be re-examined and modified to account for the crucial role of dynamin and actin in Flat $\rightarrow\Lambda$, $\Lambda\rightarrow\Omega$ and $\Omega\rightarrow\text{O}$ transitions.”

At the beginning of Discussion, the sentence “The present work revealed unexpected yet crucial mechanical roles of F-actin and dynamin ... ” was modified as “The present work revealed crucial mechanical roles of F-actin and dynamin” (the word “unexpected” is deleted as the reviewer suggested).

Page 19, 2nd paragraph: “This finding not only provides real-time imaging data supporting actin involvement in pit formation⁵⁻⁷, but also may explain ...” (emphasizing that our data support the early work as reviewed in Refs. 5-7).

3. How do the authors reconcile the current finding of dynamin’s role in forming flat to invaginated transitions, with the previous observations of multiple invaginations and long tubules in dynamin knock-out cells? These well-known older electron microscopy observations were interpreted as endocytic structures that failed to be excised, thus suggesting that dynamin is not essential for pit formation.

Reply: As the reviewer suggested, we included a paragraph discussing two reasons that may reconcile our live-cell observations with previous EM observations as in the following.

Page 22, last paragraph – page 23, first paragraph: “Dynamin knockout leads to accumulation of Ω -profiles and long tubes at the plasma membrane⁴⁶⁻⁴⁸. We consider two reasons that may reconcile this EM observation with our live-cell observation that dynamin is involved in not only $\Omega \rightarrow O$, but also $Flat \rightarrow \Lambda$ and $\Lambda \rightarrow \Omega$. First, the EM observation was obtained from cells lacking dynamin for days to weeks⁴⁶⁻⁴⁸, whereas $Flat \rightarrow \Lambda$, $\Lambda \rightarrow \Omega$ and $\Omega \rightarrow O$ transitions were measured within 60 s after $depol_{1s}$ in live cells (Figs. 3, 4, 6). If dynamin inhibition prolongs $Flat \rightarrow \Lambda$ and $\Lambda \rightarrow \Omega$ transition time from seconds to minutes or hours, but not days or weeks, it may explain why dynamin inhibition reduced $Flat \rightarrow \Lambda$ and $\Lambda \rightarrow \Omega$ transitions measured within 60 s (Figs. 3, 4, 6), but increased the number of invaginations observed with EM in days or weeks afterwards⁴⁶⁻⁴⁸. Similarly, if dynamin inhibition reduces $Flat \rightarrow \Lambda$ and $\Lambda \rightarrow \Omega$ transition by a lesser extent than $\Omega \rightarrow O$ transition measured in days or weeks after the transition is initiated, accumulation of Ω -profiles by dynamin inhibition would be observed in days or weeks later. Second, Λ -profiles have not been quantified in cells lacking dynamin⁴⁶⁻⁴⁸, likely because without real-time observation, it is difficult to determine whether a Λ -profile stems from an endocytic $Flat \rightarrow \Lambda$ transition or from a non-endocytic membrane folding. Furthermore, without real-time observation, a change in the Λ -profile number reflects the net change of both $Flat \rightarrow \Lambda$ and $\Lambda \rightarrow \Omega$, making it difficult to pinpoint which specific transition is affected. Accordingly, it remains unclear whether dynamin knockout affects Λ -profile formation or not. Our finding of dynamin involvement in $Flat \rightarrow \Lambda$, $\Lambda \rightarrow \Omega$ and $\Omega \rightarrow O$ is thus not in conflict with the observation of accumulated invaginations in dynamin knockout cells.”

4. The authors do a great job at quantifying the occurrence of all the transitions studied, this is very valuable since quantitative microscopy is sometimes overlooked in the field. The transitions from Λ to Ω and from Ω to O have a very low probability (0.12 and 0.24, respectively). What is the interpretation of this finding? What happens in the remaining ~0.8 cases, the pit collapses back to flat and there is no endocytosis? What is the biological relevance of having such a stochastic mechanism for endocytosis?

Reply: We appreciate the reviewer’s support of our quantification of each endocytic transition. As the reviewer indicated, low $Prob(\Lambda \rightarrow \Omega)$ (probability of Λ to undergo $\Lambda \rightarrow \Omega$) and $Prob(\Omega \rightarrow O)$ (probability of Ω to undergo $\Omega \rightarrow O$) often prevented $Flat \rightarrow \Lambda$ and $\Lambda \rightarrow \Omega$ from reaching O , leaving many newly generated Λ and Ω to stay unchanged within our 60 s recording time (see Figure 4A in Shin W. et al., Neuron, 2021), during which their reverse to $Flat$ or Λ was negligible (see Fig. S4 in Shin W. et al., 2021). The newly generated Λ and Ω may have to wait for next depolarization to make their transitions towards O , although very slow transition

towards O beyond our 60 s imaging time could not be excluded (Shin et al., 2021). Upon depolarization that induces exocytosis, vesicle reformation primarily comes from preformed- $\Omega \rightarrow O$, but not from flat-to-round transformation as generally believed (due to the low $\text{Prob}(\Lambda \rightarrow \Omega)$) (Shin et al., 2021). Preformed- $\Omega \rightarrow O$ transition, together with kiss-and-run, mediates diverse modes of endocytosis, such as ultrafast, fast, slow, overshoot, and bulk endocytosis (Shin et al., 2021). Thus, without the need of undergoing flat-to- Ω transition, depolarization-induced preformed- $\Omega \rightarrow O$ readily explains ultrafast (tens of milliseconds) and fast (a few seconds or less) endocytosis that seems difficult to accomplish if one assumes an endocytic flat-to-round transformation after depolarization (Shin et al., 2021).

It is unclear why $\text{Prob}(\Lambda \rightarrow \Omega)$ (~ 0.12) and $\text{Prob}(\Omega \rightarrow O)$ (~ 0.24) are low. We speculate that low $\text{Prob}(\Lambda \rightarrow \Omega)$ and $\text{Prob}(\Omega \rightarrow O)$ might be due to difficulty of significant shape changes mediated by dynamin oligomerization that may surround and constrict large Λ 's base and Ω 's pore. Given that Λ 's base are larger than Ω 's pore, Λ 's base constriction might take more energy than Ω 's pore constriction, which might explain why $\text{Prob}(\Lambda \rightarrow \Omega)$ (~ 0.12) is lower than $\text{Prob}(\Omega \rightarrow O)$ (~ 0.24).

In response to the reviewer's questions about physiological relevance and potential mechanisms of low probabilities of transition, we included a paragraph summarizing the above discussion in the revised manuscript as in the following.

Page 22, last paragraph – page 23, first paragraph : “ $\text{Prob}(\Lambda \rightarrow \Omega)$ and $\text{Prob}(\Omega \rightarrow O)$ are low, ~ 0.12 and 0.24 , respectively (Figs. 4, 6), leaving many newly generated Λ and Ω to stay at the plasma membrane (without reversing back to flat membrane)¹⁸. Upon next depolarization, the low $\text{Prob}(\Lambda \rightarrow \Omega)$ makes preformed- $\Omega \rightarrow O$ the main transition for vesicle reformation, which, together with kiss-and-run, mediates diverse endocytic modes, including ultrafast, fast, slow, overshoot, and bulk endocytosis¹⁸. The low $\text{Prob}(\Lambda \rightarrow \Omega)$ and $\text{Prob}(\Omega \rightarrow O)$ might be due to difficulty of significant shape changes mediated by dynamin oligomerization that may surround and constrict large Λ 's base and Ω 's pore. Considering that Λ 's base are larger than Ω 's pore, Λ 's base constriction might take more energy than Ω 's pore constriction, which might explain why $\text{Prob}(\Lambda \rightarrow \Omega)$ (~ 0.12) is lower than $\text{Prob}(\Omega \rightarrow O)$ (~ 0.24).”

Reviewer #3

In this manuscript, Shin, Zucker and Kundu et al showed the role of dynamin and filamentous actin on membrane budding process using STED imaging. Overall, it is a very well-written manuscript with well-performed experiments. In my opinion, this paper should be published in Nature Communications. I mostly evaluated the imaging and membrane aspect of the manuscript. My minor comments are below.

Reply: We appreciate the reviewer's strong support of our work.

Do the authors have control experiments where the budding cannot occur (e.g. no Calcium). This should rule out STED-induced tubulation artefacts.

Reply: Yes, we do have these control experiments. In a recent study (Shin W. et al., Neuron, 2021), we presented three sets of evidence suggesting that calcium influx triggers every endocytic transition, including Flat $\rightarrow\Lambda$, $\Lambda\rightarrow\Omega$ and $\Omega\rightarrow O$. First, depol_{1s} triggers Flat $\rightarrow\Lambda$, $\Lambda\rightarrow\Omega$ and $\Omega\rightarrow O$, because these transitions were observed mostly within ~ 10 s after depol_{1s} , and were rarely observed without depol_{1s} (Fig. 4D-F in Shin W. et al., 2021). Second, $N_{\text{Flat}\rightarrow\Lambda}$ (the number

of Flat $\rightarrow\Lambda$ per 10 μm membrane along the X-axis), Prob($\Lambda\rightarrow\Omega$) (probability of Λ to undergo $\Lambda\rightarrow\Omega$) and Prob($\Omega\rightarrow\text{O}$) (probability of Ω to undergo $\Omega\rightarrow\text{O}$) measured after depol_{1s} were much higher in cells with a calcium current (ICa) > 300 pA (1233 ± 48 pA, 337 cells) than with ICa < 300 pA (155 ± 6 pA, 176 cells, Fig. 4G in Shin W. et al., 2021). Although 300 pA was arbitrarily set, adjusting to 200 or 400 pA yielded similar results. Third, replacing bath calcium with strontium nearly abolished Flat $\rightarrow\Lambda$, $\Lambda\rightarrow\Omega$ and $\Omega\rightarrow\text{O}$, but did not affect capacitance jump or current amplitude (58 cells, Fig. 4H in Shin W. et al., 2021). Consequently, strontium application substantially reduced N_{Flat $\rightarrow\Lambda$} , Prob($\Lambda\rightarrow\Omega$) and Prob($\Omega\rightarrow\text{O}$) (Fig. 4H in Shin W. et al., 2021). These results indicate that Flat $\rightarrow\Lambda$, $\Lambda\rightarrow\Omega$ and $\Omega\rightarrow\text{O}$ are not STED-induced tubulation artifacts, but are triggered by calcium influx.

In response to the reviewer's question, we included a paragraph summarizing the above discussion in the revised manuscript as in the following.

Page 33, 2nd paragraph: "N_{Flat $\rightarrow\Lambda$} , Prob($\Lambda\rightarrow\Omega$) and Prob($\Omega\rightarrow\text{O}$) were negligible when measured in the absence of depol_{1s} (Fig. 4 in Ref. ¹⁸). N_{Flat $\rightarrow\Lambda$} , Prob($\Lambda\rightarrow\Omega$) and Prob($\Omega\rightarrow\text{O}$) measured after depol_{1s} were much higher in cells with ICa > 300 pA than with ICa < 300 pA (Fig. 4G in Ref. ¹⁸). Furthermore, N_{Flat $\rightarrow\Lambda$} , Prob($\Lambda\rightarrow\Omega$) and Prob($\Omega\rightarrow\text{O}$) measured after depol_{1s} were substantially reduced when extracellular calcium was replaced with strontium (Fig. 4H in Ref. ¹⁸). These results indicate that Flat $\rightarrow\Lambda$, $\Lambda\rightarrow\Omega$ and $\Omega\rightarrow\text{O}$ are triggered by calcium influx, excluding the possibility that they are STED-laser-induced artifacts."

Do authors have control experiments ruling out FRET between different fluorophores that can bias the results on protein distributions.

Reply: We labelled membrane with PH-mNeonGreen (PH-mNG, excited at 511 nm, fluorescence collected at 516-587 nm), but labelled F-actin and dynamin with lifeact-mTFP1 and dynamin 2-mTFP1 (excited at 442 nm, fluorescence collected at 447-505 nm). If fluorescence resonance energy transfer (FRET) between mTFP1 and mNG takes place, the admitted fluorescence of mTFP1 is absorbed by nearby mNeonGreen within ~10 nm. Lifeact-mTFP1 and dynamin 2-mTFP1 fluorescence could be underestimated, but the observed mTFP1 fluorescence should reflect their minimal distribution. Hence, our conclusion that F-actin and dynamin are physically available near the membrane to mediate pulling and constriction should not be affected by the potential FRET between mNG and mTFP1.

The following evidence, including new data we collected in response to this comment (new imaging with newly generated fluorescent probes), suggest that the observed distribution of lifeact-mTFP1 and dynamin 2-mTFP1 is not significantly affected by FRET.

For lifeact-mTFP1 labelling, our main findings are that lifeact-mTFP1-labelled filamentous F-actin grew at the tip of PH-mNG-labelled Λ -profile (e.g., Fig. 1f, 1g) and that F-actin surrounded Λ -profile's base (e.g., Fig. 3f). In both cases, lifeact-mTFP1-labelled F-actin did not overlap with PH-mNG-labelled plasma membrane: most mTFP1-labelled actin filaments was above the tip of the mNG-labelled Λ -profile (e.g., Fig. 1f, 1g), and mTFP1-labelled F-actin ring was more than 60 nm periphery of mNG-labelled Λ -profile's base (e.g., Fig. 3f). Thus, our main finding about F-actin distribution should not be influenced by the FRET.

For dynamin 2-mTFP1 labelling, our main findings are that dynamin 2-mTFP1 is located above the tip of the mNG-labelled Λ -profile (e.g., Figs. 3b, 4c), at the periphery of Λ -profile's base (Figs. 3d, 4c), and at the pore region of the Ω -profile (e.g., Fig. 4e). Since dynamin 2-mTFP1 puncta is above the tip of the mNG-labelled Λ -profile (e.g., Fig. 3b), there should be no

FRET from mNG to influence the observation of mTFP1. Since dynamin 2-mTFP1 puncta are in general not overlapped with strong mNG-labelled Λ -profile's base (e.g., Fig. 3d), the observation that dynamin-mTFP1 surrounds Λ -profile's base should not be significantly influenced by FRET from strong nearby (<10 nm) mNG.

For dynamin 2-mTFP1 puncta at the mNG-labelled Ω -profile's pore, we performed new experiments by imaging dynamin 2-mNG and PH-mTFP1, in which dynamin 2-mNG fluorescence cannot be absorbed by nearby PH-mTFP1, and thus should not be affected by FRET. We found that dynamin 2-mNG was associated with PH-mTFP1-labelled Ω -profile's pore region (n = 10 events, Supplementary Fig. 8a), similar to the association of dynamin 2-mTFP1 with PH-mNG-labelled Ω -profile's pore region (n = 15, Supplementary Fig. 8b). We also observed that dynamin 2-mNG was associated with PH-mTFP1-labelled Λ -profile (n = 9 events, Supplementary Fig. 8c), similar to the association of dynamin 2-mTFP1 with PH-mNG-labelled Λ -profile (e.g., Figs. 3b, 3d). We concluded that FRET does not significantly influence our conclusion that F-actin and dynamin are physically available near Λ - and Ω -profile to generate pulling and constriction forces.

In the revised manuscript, we included the above discussion at page 31 (last two paragraphs) – page 32 (first three paragraphs), which should address the reviewer's concern.

Laser powers used for STED imaging should be given as absolute numbers (microwatt, milliwatt), not as % of the total powers where the total powers are not mentioned. xz images might be interpreted as either 3D image with 2D-STED or 3D-STED. It should clearly be mentioned whether the authors have done 2D STED using 2D doughnut, or z(3D)-STED using 3D depletion pattern.

Reply: As the reviewer suggested, we provided the laser powers for each imaging condition in the revised manuscript as in the following.

Page 32, 4th-5th paragraphs: “For three-color STED imaging with 592 nm STED depletion laser, dynamin 2-mTFP1 (or dynamin 1-mTFP1 or Lifeact-mTFP1), PH_G and A532 were excited at 442 nm (power: 1-3 mW), 507 nm (power: 1-5 mW), and 545 nm (power: 4-6 mW), respectively, and their fluorescence collected at 447-502 nm, 512-540 nm, and 550-587 nm, respectively. For two-color STED imaging of dynamin 2-mTFP1 (or dynamin 1-mTFP1 or Lifeact-mTFP1) and PH_G with the 592 nm STED depletion beam, mTFP1 and PH_G were sequentially excited at 442 nm (power: 1-3 mW) and 511 nm (power: 1-5 mW), respectively, and their fluorescence collected at 447-505 nm and 516-587, respectively. STED XZ-plane imaging was performed with 3D depletion pattern [z(3D)-STED]. The depletion laser power distribution of depletion 3D doughnut was 60% or higher in Z direction and 40% or lower in XY direction. The depletion laser power was 100-500 mW for imaging of PH_G, dynamin 2-mTFP1 or lifeact-mTFP1, but 100-200 mW for imaging of A532.

The excitation laser power for A532 was 4-6 mW, at which fluorescent A532 can be bleached within a few seconds during XZ/Y_{fix} imaging every 26-200 ms.”

Red/green image combination is impossible to read for color-blind readers, I would consider changing red to magenta.

Reply: We respect the reviewer's point. We did use magenta color in some plots that require three colors or more. Therefore, even if we change red to magenta, we still have to use red color in some plots. Red/green combination provides the sharpest color contrast, and thus has been

used in many papers, including most of our previous papers. We hope to continue using this sharpest color contrast and thus would not change red to magenta.

REVIEWERS' COMMENTS

Reviewer #2 (Remarks to the Author):

The authors have adequately addressed the points raised by the reviewers, I appreciate the explanations and clarifications given to my questions. I think the manuscript is ready for publication and no further work is needed.

Reviewer #3 (Remarks to the Author):

Authors addressed all my concerns. Therefore, I recommend the publication.

Reviewer: Erdinc Sezgin

Point-to-point response

The reviewers did not raise any technical issues in this round of review. Accordingly, we do not have any point-to-point response.